# Top-N: Equivariant set and graph generation without exchangeability

**Clément Vignac, Pascal Frossard**
LTS4, EPFL
Lausanne, Switzerland

## Abstract

This work addresses one-shot set and graph generation, and, more specifically, the parametrization of probabilistic decoders that map a vector-shaped prior to a distribution over sets or graphs. Sets and graphs are most commonly generated by first sampling points i.i.d. from a normal distribution, and then processing these points along with the prior vector using Transformer layers or Graph Neural Networks. This architecture is designed to generate exchangeable distributions, i.e., all permutations of the generated outputs are equally likely. We however show that it only optimizes a proxy to the evidence lower bound, which makes it hard to train. We then study equivariance in generative settings and show that non-exchangeable methods can still achieve permutation equivariance. Using this result, we introduce *Top-n creation*, a differentiable generation mechanism that uses the latent vector to select the most relevant points from a trainable reference set. Top-n can replace i.i.d. generation in any Variational Autoencoder or Generative Adversarial Network. Experimentally, our method outperforms i.i.d. generation by 15% at SetMNIST reconstruction, by 33% at object detection on CLEVR, generates sets that are 74% closer to the true distribution on a synthetic molecule-like dataset, and generates more valid molecules on QM9.

## 1 Introduction

In recent years, many architectures have been proposed for learning vector representations of sets and graphs. Sets and graphs are unordered and therefore have the same symmetry group, which offers a fruitful theoretical framework for the development of new models (Bronstein et al., 2021). The opposite problem, i.e., learning to generate sets and graphs from a low-dimensional vector prior, has been less explored. Yet, it has important applications in drug discovery, for example, where neural networks can learn to sample stable drug-like molecules conditioned on some molecule-level properties such as solubility or synthesizability.

Set and graph generation are very similar problems, and one can navigate between set and graph representations using Set2Graph functions (Serviansky et al., 2020) or graph neural networks. We therefore generally treat them together in this paper, but only use the terminology of sets to avoid overly abstract notations. The distinctive properties of graph generation (e.g., graph matching problems) are discussed when needed.

There are two main classes of probabilistic decoders for sets: recursive and one-shot. Recursive generators are conceptually simpler, since they add points one by one (Liu et al., 2018; Liao et al., 2019; Sun et al., 2020; Nash et al., 2020). They are however slow and introduce an order in the points that does not exist in the data. In this work, we instead focus on one-shot generation, which allows for more principled designs. By definition, one-shot models feature a layer that maps a vector to an initial set. We group all layers until this one into a *creation module*, and the subsequent layers into an *update module*. The design of the update is well understood: as it maps a set to another set, any permutation equivariant network suits this task. In contrast, the creation step poses two major challenges, which are i) the need to generate sets of various sizes, and ii) the generation of different points from a single prior vector, which cannot be done with a permutation equivariant function.

Set creation is typically performed by first sampling points independently from a normal distribution, and then appending the latent vector to each point. This design allows the generation of any number

of points and decouples the set cardinality from the number of trainable parameters. Furthermore, it generates exchangeable distributions (all permutations of a set are equally likely), a property which is commonly held as the equivalent of equivariance for generative models. However, it was empirically observed that VAEs based on independent sampling are hard to train, which results in limited performance (Krawczuk et al., 2021).

In this work, we first propose a theoretical argument to this empirical observation, by showing that VAEs that use independent sampling only optimize a proxy to the evidence lower bound (ELBO). As the standard definition of equivariance cannot be used in generative settings, we then propose a generalization of this notion called $(F, l)$-equivariance: informally, an architecture is $(F, l)$-equivariant if the parameter updates do not depend on the group elements used to represent training data. We derive sufficient conditions for equivariance in this setting. They reveal that $(F, l)$-equivariance explains the loss functions commonly used both in generative and discriminative tasks, and suggest that exchangeability may not be useful in GANs and VAEs.

Based on these results, we finally propose a non-exchangeable set creation method called *Top-n creation*. Our method relies on a trainable reference set where each point $i$ has a *representation* $r_i \in \mathbb{R}^c$ and an *angle* $\phi_i \in \mathbb{R}^a$. To generate a set with $n$ elements, we select the $n$ points whose angles have the largest cosine with the latent vector. In order to make this process differentiable, we build upon the Top-$K$ pooling mechanism initially proposed for graph coarsening (Gao & Ji, 2019). Top-n can eventually be integrated in any VAE or GAN to form a complete generative model for sets or graphs. Our method is easier to train than stochastic generators, and has better generalization performance than other existing methods.

We benchmark Top-n on both set and graph generation tasks: it is able to reconstruct the data more accurately on a set version of MNIST, generalize better on the CLEVR object detection dataset, fit more closely the true distribution on a dataset of synthetic molecule-like structures in 3D, and generate realistic molecular graphs on QM9.

## 2 THE ONE-SHOT SET GENERATION PROBLEM

We consider the problem of learning a probabilistic decoder $f$ that maps latent vectors $z \in \mathbb{R}^l$ to multi-sets[1] $\mathcal{X} = \{x_1, ..., x_n\}$ that contain a varying number $n$ of points $x_i$ in $\mathbb{R}^d$. Given sample sets from an unknown distribution $\mathcal{D}$, $f$ should be such that, if $z$ is drawn from a prior distribution $p_Z(z)$, then the push-forward measure $f_\#(p_Z)$ (i.e., the law of $f(z)$) is close to $\mathcal{D}$. In practice, representing sets is not convenient on standard hardware, and sets are internally represented by matrices $X \in \mathbb{X} = \bigcup_{n \in \mathbb{N}} \mathbb{R}^{n \times d}$ where each row represents a point $x_i \in \mathbb{R}^d$. Algorithms that return a set implicitly assume the use of a function $\mathrm{mat\text{-}to\text{-}set}$ that maps $X$ to the corresponding set $\mathcal{X}$.

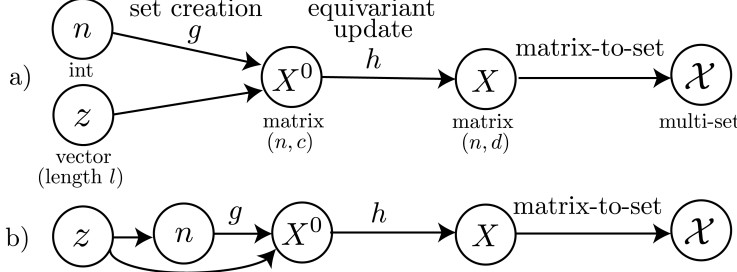

Figure 1: The graphical models for set generation. The number of points can either be sampled from the dataset distribution (a), or learned from the latent vector (b). While any equivariant function can be used for the update $h$, the set creation $g$ concentrates the challenges of set generation. For graph generation, edge weights are generated in addition to the node features matrices $X^0$ and $X$.

Existing architectures for one-shot generation use one of the graphical models described in Figure 1. First, a number of points for the set has to be sampled. Most works assume that the set cardinalities are known during training. At generation time, they sample $n$ from the distribution of set

---

[1]For the sake of simplicity, we will refer to *sets* instead of *multi-sets* in the rest of the paper.

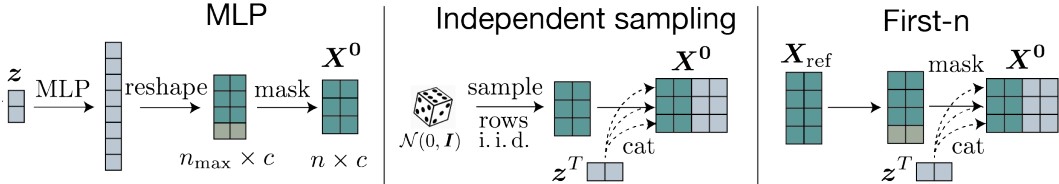

Figure 2: Existing creation methods for mapping a latent vector $\boldsymbol{z}$ to a set of points $\boldsymbol{X}^0$. First-n creation empirically gives the best performance. It learns a reference set represented by a matrix $\boldsymbol{X}_{\text{ref}}$, and concatenates the latent vector to each point of this set.

cardinalities in the training data. This method assumes that the latent vector $\boldsymbol{z}$ is independent of the number of points $n$, so that the generative mechanism writes $p(\mathcal{X}|n, \boldsymbol{z})\, p(n)\, p(\boldsymbol{z})$.

Kosiorek et al. (2020) instead propose to learn the value of $n$ from the latent vector using a MLP. This layer is trained via an auxiliary loss, but the predicted value is used only at generation time (the ground truth cardinality is used during training). The generative model is in this case $p(\mathcal{X}|\boldsymbol{z}, n)\, p(n|\boldsymbol{z})\, p(\boldsymbol{z})$, i.e., $\boldsymbol{z}$ and $n$ are not independent anymore.

Once $n$ is sampled, one-shot generation models can formally be decomposed into several components: a first function $g$ (that we call *creation function*) maps the latent vector to an initial set $\boldsymbol{X}^0 \in \mathbb{R}^{n \times c}$. This function is usually simple, and is therefore not able to model complex dependencies within each set. For this reason, $\boldsymbol{X}^0$ may then be refined by a second function $h$ that we call *update*, so that the whole model can be written $f = \text{mat-to-set} \circ h \circ g$. We now review the parametrizations that have been proposed for the creation and update modules.

## 2.1 METHODS FOR SET CREATION

**MLP based**  Many existing methods (Achlioptas et al. (2018); Zhang et al. (2020; 2021b) for sets, Guarino et al. (2017); De Cao & Kipf (2018); Simonovsky & Komodakis (2018) for graphs) learn a MLP from $\mathbb{R}^l$ to $\mathbb{R}^{n_{\max}c}$, where $n_{\max}$ is the largest set size in the training data. The output vector is then reshaped as a $n_{\max} \times c$ matrix, and masked to keep only the first $n$ rows (Figure 2). MLPs ignore the symmetries of the problem and are only trained to generate up to $n_{\max}$ points, with no ability to extrapolate to larger sets. Despite these limitations, they often perform on par with more complex methods for small graphs (Madhawa et al., 2020; Mitton et al., 2021). We explain in Section 3 this surprising phenomenon.

**Independent sampling**  Along with MLP based generation, the most popular method for set and graph creation is to draw $n$ points i.i.d. from a low dimensional normal distribution, and to concatenate the latent vector to each sample (Köhler et al., 2020; Yang et al., 2019b; Kosiorek et al., 2020; Stelzner et al., 2020; Satorras et al., 2021; Zhang et al., 2021a; Liu et al., 2021). The main advantage of independent sampling is that it does not constrain the number of points that can be generated. Furthermore, it is exchangeable, i.e., all permutations of the rows of $\boldsymbol{X}^0$ are equally likely. This property is widely considered as the equivalent of equivariance for generative models (Yang et al., 2019a; Biloš & Günnemann, 2021; Kim et al., 2021; Köhler et al., 2020; Li et al., 2020).

However, it was empirically observed that VAEs built with a i.i.d. creation mechanism fail to fit the training data correctly (Krawczuk et al., 2021), which reflects in the poor quality of the sampled sets. To understand why, consider a VAE made of an encoder $q_\phi(\boldsymbol{z}|\boldsymbol{X})$ and a decoder $f_\theta(\boldsymbol{X}|\boldsymbol{z})$ parametrized by $\phi$ and $\theta$. Variational autoencoders maximize the evidence lower bound (ELBO) $\mathcal{L}$, which is a proxy for the data likelihood under the model:

$$\mathcal{L}(\boldsymbol{X}) = \mathbb{E}_{q_\phi(\boldsymbol{z}|\boldsymbol{X})}[\log p_\theta(\boldsymbol{X}, \boldsymbol{z}) - \log q_\phi(\boldsymbol{z}|\boldsymbol{X})] \leq p_\theta(\boldsymbol{X}) \tag{1}$$

Probabilistic decoders $f_\theta(\boldsymbol{z})$ based on independent sampling are stochastic, so that $\log p_\theta(\boldsymbol{X}, \boldsymbol{z})$ cannot be computed in close form. By conditioning on the initial set $\boldsymbol{X}^0$ and using Jensen's inequality, we have:

$$\log p_\theta(\boldsymbol{X}, \boldsymbol{z}) = \log \mathbb{E}_{\boldsymbol{X}^0 \sim p(\boldsymbol{X}^0)}\, p_\theta(\boldsymbol{X}, \boldsymbol{z}|\boldsymbol{X}^0) \geq \mathbb{E}_{\boldsymbol{X}^0 \sim p(\boldsymbol{X}^0)} \log p_\theta(\boldsymbol{X}, \boldsymbol{z}|\boldsymbol{X}^0) \tag{2}$$

which gives in expectation

$$\mathcal{L}(\boldsymbol{X}) \geq \mathbb{E}_{\boldsymbol{X}^0, \boldsymbol{z}}\left[\log p_\theta(\boldsymbol{X}, \boldsymbol{z}|\boldsymbol{X}^0) - \log q_\phi(\boldsymbol{z}|\boldsymbol{X})\right] := \mathcal{L}'(\boldsymbol{X}). \tag{3}$$

Methods based on independent sampling use a Monte-Carlo estimate for $\nabla_{\theta,\phi}\mathcal{L}'(\boldsymbol{X})$ that leverages the reparametrization trick. They therefore only optimize a lower bound of the ELBO, which could explain why they are difficult to train.

**First-n**   Instead of sampling points, Zhang et al. (2019) and Krawczuk et al. (2021) propose to always start from the same learnable set $\boldsymbol{X}_{\mathrm{ref}} \in \mathbb{R}^{n_{\max} \times c}$, and mask this matrix to keep only the first $n$ rows: we therefore call this method First-n creation. Similarly to the independent sampling method, the latent vector is then concatenated to each point of this set.

Empirically, First-n converges much faster than sampling-based methods (Krawczuk et al., 2021), but the network is only trained to generate up to $n_{\max}$ points and has no ability to extrapolate to larger sets. Furthermore, selecting the first $n$ rows of the reference set $\boldsymbol{X}_{\mathrm{ref}}$ introduces a bias because the first rows are selected more often than the last ones.

**Creation methods for graph generation**   For graph generation, edge weights (or edge features) also need to be learned. To the best of our knowledge, First-n creation has not been used for this purpose yet, and the weights are generated either by a MLP or by sampling normal entries i.i.d. An alternative is to first generate a set, and then use a Set2Graph update function in order to learn the graph adjacency matrix as in (Bresson & Laurent, 2019; Krawczuk et al., 2021).

## 2.2   METHODS FOR SET UPDATE

Since all creation methods except MLPs can only generate very simple sets and adjacency matrices, additional layers are usually used to refine these objects – we gather these layers into the *update* module. Because these layers map a set or a graph to another set/graph, the update module falls into the standard framework of permutation equivariant representation learning and all existing equivariant layers can be used: Deep sets (Zaheer et al., 2017), self-attention (Vaswani et al., 2017; Lee et al., 2019), Set2Graph (Serviansky et al., 2020), graph neural networks (Battaglia et al., 2018) or higher-order neural networks (Morris et al., 2019). Recently, Transformer layers have constituted the most popular method for set and graph update (Bresson & Laurent, 2019; Kosiorek et al., 2020; Stelzner et al., 2020; Krawczuk et al., 2021) – we also use such layers in our experiments.

## 3   A PERMUTATION EQUIVARIANCE VIEW ON SET GENERATION

Whereas exchangeability is usually considered as a key feature of independent sampling, we have seen that this method empirically does not outperform other strategies. To understand why, we need to study equivariance in generative models and propose a relevant definition in this setting.

As set and graphs are unordered, the symmetric group $\mathbb{S} = \bigcup_{n \in \mathbb{N}^*} \mathbb{S}_n$ containing all permutations is a symmetry of these tasks. A permutation $\pi \in \mathbb{S}_n$ acts on a $n \times n$ matrix $\boldsymbol{A}$ by permuting its rows and columns (which we write $\pi.\boldsymbol{A} = \pi \, \boldsymbol{A} \, \pi^T$), on a $n \times c$ matrix by permuting its rows ($\pi.\boldsymbol{X} = \pi\boldsymbol{X}$), and leaves a vector $\boldsymbol{z} \in \mathbb{R}^h$ unchanged ($\pi.\boldsymbol{z} = \boldsymbol{z}$). This symmetry constitutes a useless factor of variation in the data that should be factored out in the latent space (i.e., $\pi.\boldsymbol{z} = \boldsymbol{z}$).

In discriminative models, symmetries are accounted for when a neural network $f$ is equivariant to the action of a group, which writes $\pi.f(\boldsymbol{X}) = f(\pi.\boldsymbol{X})$ (Kondor, 2008). When the input of $f$ is a vector, imposing $\pi.f(\boldsymbol{z}) = f(\pi.\boldsymbol{z}) = f(\boldsymbol{z})$ however only allows for solutions where all rows are equal, which is too restrictive. To solve this issue, we propose a definition called $(F, l)$-equivariance which generalizes the common one, but provides more relaxed conditions in generative settings.

Our proposition is based on the assumption that the main role of equivariance is to make data augmentation useless. In discriminative settings, this is normally done by combining an equivariant model with an invariant loss function. For example, the $l_2$ loss is commonly used to learn the future state of a $n$-body system, but not the $l_1$ loss, as it is not rotation invariant. Formally, if $F_\Theta = \{f_\theta : \mathbb{X} \to \mathbb{Y}; \ \theta \in \Theta\}$ is an hypothesis class of $\mathbb{G}$-equivariant functions from $\mathbb{X}$ to $\mathbb{Y}$ (for example a neural architecture parametrized by $\theta$), then the loss functions $l$ should satisfy

$$\forall f \in F, \ \forall g \in \mathbb{G}, \ \forall(\boldsymbol{X}, \boldsymbol{Y}) \in \mathbb{X} \times \mathbb{Y}, \qquad l(g.f(\boldsymbol{X}), g.\boldsymbol{Y}) = l(f(\boldsymbol{X}), \boldsymbol{Y}) \qquad (4)$$

Furthermore, we observe that when $l$ satisfies Eq. 4, the gradients with respect to the parameters satisfy $\nabla_\theta \, l(f(g.\boldsymbol{X}), g.\boldsymbol{Y}) = \nabla_\theta \, l(f(\boldsymbol{X}), \boldsymbol{Y})$, i.e., each parameter update is independent of the

group elements that are used to represent $\boldsymbol{X}$ and $\boldsymbol{Y}$. It follows that the training dynamics as a whole become independent of the group elements used to represent the data. We propose to use this property to define equivariance:

**Definition 1** $((F, l)$-equivariance). Consider an hypothesis class $F_\Theta \subset \mathbb{Y}^{\mathbb{X}}$, a group $\mathbb{G}$ that acts on $\mathbb{X}$ and $\mathbb{Y}$ and a loss function $l$ defined on $\mathbb{Y}$. We say that the pair $(F_\Theta, l)$ is equivariant to the action of $\mathbb{G}$ if the dynamics of $\theta \in \Theta$ trained with gradient descent on $l$ do not depend on the group elements that are used to represent the training data.

By construction, using an equivariant architecture and an invariant loss is sufficient for $(F, l)$-equivariance in discriminative settings. For standard generative architectures for sets and graphs, we derive the following sufficient conditions (proofs are given in Appendix A):

**Lemma 1.** *Sufficient conditions for $(F, l)$-equivariance:*

1. *GANs: if $F$ is a GAN architecture with a permutation invariant discriminator, and $l$ the standard GAN loss, then $(F, l)$ is permutation equivariant. No constraint is imposed on the generator.*
2. *VAEs: if $F$ is an encoder-decoder architecture with a permutation invariant encoder, and the reconstruction loss $l$ satisfies $\forall \pi \in \mathbb{S}, l(\pi.\boldsymbol{X}, \hat{\boldsymbol{X}}) = l(\boldsymbol{X}, \hat{\boldsymbol{X}})$, then $(F, l)$ is permutation equivariant. No constraint is imposed on the decoder function.*
3. *Normalizing flows: if $F$ is an architecture such that the set creation yields an exchangeable distribution, the update is permutation equivariant and invertible, and $p_\theta$ denotes the model likelihood, then $(F, -\log p_\theta)$ is permutation equivariant (proved in Köhler et al. (2020)).*

As desired, $(F, l)$-equivariance does not impose $\pi.f(\boldsymbol{z}) = f(\pi.\boldsymbol{z})$ in generative architectures, but still makes data augmentation unnecessary. Furthermore, the constraints of Lemma 1 are satisfied by most existing architectures, including early ones (Simonovsky & Komodakis, 2018; De Cao & Kipf, 2018; Köhler et al., 2020). In particular, the constraint on the loss for VAEs is satisfied by the two loss functions commonly used for sets, namely Chamfer loss and the Wasserstein-2 distance (defined in Appendix B). Our definition, introduced by observing common practice in discriminative settings, is therefore able to explain common practice for generative tasks as well.

We finally observe that exchangeability does not appear in the sufficient conditions for GANs and VAEs. To understand why, recall that in GANs and VAEs a mat-to-set function is implicitly applied to the model output: a method that generates matrices that are always permuted in the same way is therefore equivalent to one that generates exchangeable matrices. In other words, the fact that the output of the model is not a matrix but a set is an assumption, not something that needs to be proved[2]. This observation explains why independent sampling creation does not outperform non-exchangeable set creation methods such as MLPs and First-n. In the following section, we therefore design a new creation mechanism without worrying about the model exchangeability.

## 4 THE TOP-N CREATION MECHANISM

We have seen in Section 2 that existing set creations methods suffer from important limitations: independent sampling makes it hard to train the model, while MLPs and First-n use a fixed mask to select the correct number of points and cannot extrapolate to larger sets. In order to solve these limitations, we propose a new method called Top-n creation, which is summarized in Figure 3.

Similarly to First-n, Top-n also uses a reference set, but in Top-n this set can have an arbitrary size $n_0$. Each point in this set is a pair $(\boldsymbol{\phi}, \boldsymbol{r})$ : the *angle* $\boldsymbol{\phi} \in \mathbb{R}^a$ is used to decide when to select the point, and $\boldsymbol{r} \in \mathbb{R}^c$ contains the *representation* of the point. Given a latent vector $\boldsymbol{z} \in \mathbb{R}^{l \times 1}$, a reference set made of angles $\boldsymbol{\Phi} \in \mathbb{R}^{n_0 \times a}$ and representations $\boldsymbol{R} \in \mathbb{R}^{n_0 \times c}$, as well as learnable

---

[2]On the contrary, normalizing flows cannot use the non-invertible mat-to-set function, and typically compute probability distributions on the space of matrices rather than multisets. It is therefore natural that exchangeability appears for normalizing flows and not for the other architectures.

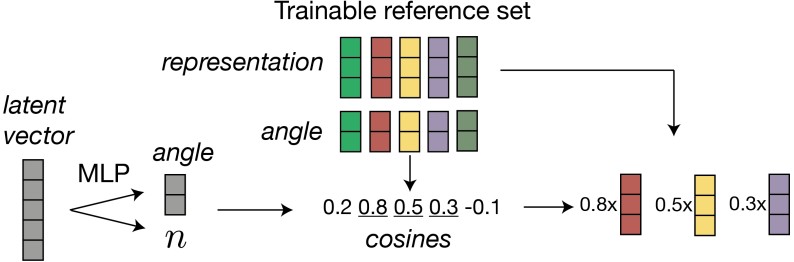

Figure 3: Top-n creation learns to select the most relevant points in a trainable reference set based on the value of the latent vector. To obtain gradients and train the angles and the MLP despite the non-differentiable argsort operation, we modulate the selected representations with the values of the cosines – in practice, we use a FiLM layer (Perez et al., 2018) rather than multiplication.

matrices $\boldsymbol{W}_1$ to $\boldsymbol{W}_4$ (respectively of sizes $1 \times c$, $1 \times c$, $l \times c$, $l \times c$), Top-n creation computes:

$$\boldsymbol{a} = \mathrm{MLP}_1(\boldsymbol{z}) \qquad\qquad \in \mathbb{R}^a \qquad (5)$$

$$\boldsymbol{c} = \boldsymbol{\Phi}\,\boldsymbol{a}\ /\ \mathrm{vec}((\|\boldsymbol{\phi}_i\|_2)_{1 \leq i \leq n_0}) \qquad\qquad \in \mathbb{R}^{n_0} \qquad (6)$$

$$\boldsymbol{s} = \mathrm{argsort}_\downarrow(\boldsymbol{c})[:n] \qquad\qquad \in \mathbb{N}^n \qquad (7)$$

$$\tilde{\boldsymbol{c}} = \mathrm{softmax}(\boldsymbol{c}[\boldsymbol{s}]) \qquad\qquad \in \mathbb{R}^{n \times 1} \qquad (8)$$

$$\boldsymbol{X}^0 = \boldsymbol{R}[\boldsymbol{s}] \odot \tilde{\boldsymbol{c}}\,\boldsymbol{W}_1 + \tilde{\boldsymbol{c}}\,\boldsymbol{W}_2 \qquad\qquad \in \mathbb{R}^{n \times c} \qquad (9)$$

$$\boldsymbol{X}^0 = \boldsymbol{X}^0 \odot \boldsymbol{1}_n\,\boldsymbol{z}^T\,\boldsymbol{W}_3 + \boldsymbol{1}_n\,\boldsymbol{z}^T\,\boldsymbol{W}_4 \qquad\qquad \in \mathbb{R}^{n \times c} \qquad (10)$$

The crux of the Top-n creation module is to select the points that will be used to generate a set based on the value of the latent vector (Eq. 5, 6, 7). Unfortunately, the gradient of the $\mathrm{argsort}$ operation is 0 almost everywhere ($\partial \boldsymbol{R}[\boldsymbol{s}]/\partial \boldsymbol{\Phi} = 0$), and a mechanism has to be used in order to train the angles $\boldsymbol{\Phi}$ and the MLP of Eq. (5). In Top-n, we modulate the representation of the selected points $\boldsymbol{R}[\boldsymbol{s}]$ with the cosines $\boldsymbol{c}$ (Eq. 8). This operation provides a path in the computational graph that does not go through the $\mathrm{argsort}$, so that the gradients of $\boldsymbol{\Phi}$ and $\boldsymbol{a}$ are not always 0. For example:

$$\frac{d\boldsymbol{X}^0}{d\boldsymbol{\Phi}} = \frac{\partial \boldsymbol{X}^0}{\partial \boldsymbol{R}[\boldsymbol{s}]}\frac{d\boldsymbol{R}[\boldsymbol{s}]}{d\boldsymbol{\Phi}} + \frac{\partial \boldsymbol{X}^0}{\partial \tilde{\boldsymbol{c}}}\frac{d\tilde{\boldsymbol{c}}}{d\boldsymbol{\Phi}} = \frac{\partial \boldsymbol{X}^0}{\partial \tilde{\boldsymbol{c}}}\frac{d\tilde{\boldsymbol{c}}}{d\boldsymbol{\Phi}},$$

Equations (5) to (9) build upon the Top-K pooling mechanism used by Gao & Ji (2019) for graph coarsening, but differ in several aspects. First, Gao & Ji (2019) compute cosines between the angle $\boldsymbol{a}$ and the representations $\boldsymbol{R}$. This method tends to select points that are similar. On the contrary, we parametrize the angles and the representations independently, so that two points can have similar angles (in which case they will usually be selected together) but very diverse representations at the same time. Second, Gao & Ji (2019) uses a multiplicative modulation ($\boldsymbol{X}^0 = \boldsymbol{X}_{\mathrm{ref}}[\boldsymbol{s}] \odot \tilde{\boldsymbol{c}}$) in Eq. (9) while we use a more expressive FiLM layer (Perez et al., 2018) that combines both additive and multiplicative modulation.

Finally, while previous works usually append the latent vector to each point, we exploit the equivalence between summation and concatenation when a linear layer is applied, which writes

$$\mathrm{cat}(\boldsymbol{X}^0,\ \boldsymbol{1}_n\boldsymbol{z}^T)\,\boldsymbol{W} = \boldsymbol{X}^0\boldsymbol{W}_1 + \boldsymbol{1}_n(\boldsymbol{z}^T\boldsymbol{W}_2), \qquad (11)$$

for $\boldsymbol{W} = \mathrm{cat}(\boldsymbol{W}_1, \boldsymbol{W}_2)$. Contrary to the concatenation (left-hand side), the sum (right-hand side) does not compute $\boldsymbol{z}^T\boldsymbol{W}_2$ several times, which reduces the complexity of this layer from $O(n(c+l)c)$ to $O(nc^2 + cl + nl)$. Again, we combine the sum and multiplicative modulation in a FiLM layer to build a more expressive model in Eq. (10).

Our algorithm retains the advantages of First-n creation, but replaces the arbitrary selection of the first $n$ points by a mechanism that learns to select the most relevant points for each set. Since it also decouples the number of points in the reference set from the number of points in the training examples, the size of the reference set becomes a hyperparameter of the model. Empirically, we

Table 1: Mean Chamfer loss and 95% confidence interval over 6 runs. Methods in italic are those used in the original papers for TSPN (Kosiorek et al., 2020) and DSPN (Zhang et al., 2019) Result differ from the original papers due to a difference in the loss computation (cf. Appendix D.1).

| Method | Set creation | Chamfer (e-5) | Method | Set creation | Chamfer (e-5) |
|--------|--------------|---------------|--------|--------------|---------------|
| TSPN | *i.i.d. sampling* | $16.42_{\pm 0.53}$ | DSPN | i.i.d. sampling | $28.56_{\pm 1.23}$ |
| | First-n | $\mathbf{15.45}_{\pm 1.41}$ | | *First-n* | $26.61_{\pm 0.54}$ |
| | Top-n | $\mathbf{14.98}_{\pm 0.59}$ | | Top-n | $\mathbf{22.59}_{\pm 1.71}$ |

observe a tradeoff: when using more points in the reference set, each point is updated less often which makes training slower; however, the model tends to avoid overfitting and generalize better. Top-n creation can be used in any GAN or VAE architecture as a replacement for other set creation methods. It is however not suited to normalizing flows, because it is based on a hard selection process which is not invertible (the value of the reference points that are not selected is not used).

Since First-n and Top-n use a fixed set of reference points, one may wonder if they restrict the model expressivity. We however show that it is not the case: used with a two-layer MLP, the First-n and Top-n modules are universal approximators over sets.

**Proposition 1** (Expressivity). *For any set size $n$, maximal norm $M$ and precision parameter $\epsilon$, there is a 2-layer row-wise MLP $f$ and reference points $\{\boldsymbol{x}_1, ..., \boldsymbol{x}_n\}$ such that, for any set $\{\boldsymbol{y}_1, ..., \boldsymbol{y}_n\}$ of points in $\mathbb{R}^d$ with $\forall i, ||\boldsymbol{y}_i|| \leq M$ there is a latent vector $\boldsymbol{z}$ of size $nd \times 1$ that satisfies:*

$$||f(\mathrm{cat}(\boldsymbol{X}, \mathbf{1}_n \boldsymbol{z}^T)) - \boldsymbol{Y}||_{\mathcal{W}_2} \leq \epsilon \tag{12}$$

*where $\mathcal{W}_2$ denotes the Wasserstein-2 distance.*

## 5 EXPERIMENTS

We compare Top-n to other set and graph creation methods on several tasks: autoencoding a set version of MNIST, detecting objects on CLEVR, generating realistic 3D structures on a synthetic molecule-like dataset, and generating varied valid molecules on the QM9 chemical dataset[3]. All training curves are available in Appendix E.

### 5.1 SET MNIST

We first perform experiments on the SetMNIST benchmark, introduced in Zhang et al. (2019). The task consists in autoencoding point clouds that are built by thresholding the pixel values in MNIST images, adding noise on the locations and normalizing the coordinates. Our goal is to show that Top-n can favorably replace other set creation methods without having to tune the rest of the architecture extensively. For this purpose, we use existing implementations of DSPN (Zhang et al., 2019) TSPN (Kosiorek et al., 2020)[4], which are respectively a sort of diffusion model and a transformer-based autoencoder. Experiment details can be found in Appendix D.1.

The results for both methods are very similar (Figure 1): the model based on independent sampling has poor performance and needs more epochs to be trained than First-n and Top-n. Top-n performs consistently better for both DSPN and TSPN, which shows that it is able to select the most relevant reference points for each set.

### 5.2 OBJECT DETECTION ON CLEVR

We further benchmark Top-n on object detection with the CLEVR dataset, made of 70k training images and 15k validation images representing simple objects. Again, we use the implementation of DSPN and the setting proposed in (Zhang et al., 2019). Two tasks are evaluated. In the first one, the goal is to predict bounding boxes in each image. In the second one, the full scene should be predicted (with the shape, color, size and material of the objects). Images are encoded using a pretrained ResNet34 architecture – the resulting vector is used as input to the set generation model.

---

[3]Source code is available at `github.com/cvignac/Top-N`

[4]github.com/LukeBolly/tf-tspn (reimplementation by someone else) and github.com/Cyanogenoid/dspn

Table 2: Bounding box prediction on CLEVR. The metric is the average precision on the test set for different intersection-over-union thresholds, computed over 6 runs (higher is better).

| Model | Generator | $AP_{50}$ | $AP_{60}$ | $AP_{70}$ | $AP_{80}$ | $AP_{90}$ |
|---|---|---|---|---|---|---|
| DSPN | MLP | $93.7_{\pm1.8}$ | $82.8_{\pm3.2}$ | $59.6_{\pm4.8}$ | $26.2_{\pm4.5}$ | $1.8_{\pm0.8}$ |
| | i.i.d. sampling | $\mathbf{97.3}_{\pm2.0}$ | $\mathbf{93.2}_{\pm3.7}$ | $\mathbf{80.6}_{\pm5.4}$ | $\mathbf{51.8}_{\pm5.5}$ | $\mathbf{11.6}_{\pm2.3}$ |
| | *First-n* | $88.2_{\pm5.1}$ | $77.1_{\pm7.3}$ | $57.3_{\pm8.2}$ | $29.0_{\pm6.1}$ | $4.0_{\pm1.3}$ |
| | Top-n | $\mathbf{97.3}_{\pm1.3}$ | $\mathbf{93.0}_{\pm2.8}$ | $\mathbf{80.8}_{\pm5.0}$ | $\mathbf{53.0}_{\pm7.0}$ | $\mathbf{12.5}_{\pm3.9}$ |

Table 3: Full scene prediction on CLEVR. The metric is the average precision on the test set, computed over 6 runs (higher is better). While MLP and i.i.d. sampling have better training metrics, Top-n generalizes much better to new images.

| Model | Generator | $AP_{10}$ | $AP_{20}$ | $AP_{50}$ | $AP_{100}$ | $AP_{\text{inf}}$ |
|---|---|---|---|---|---|---|
| DSPN | MLP | $2.7_{\pm1.4}$ | $17.9_{\pm8.6}$ | $42.1_{\pm16.8}$ | $54.5_{\pm19.4}$ | $71.2_{\pm3.0}$ |
| | i.i.d. sampling | $2.6_{\pm1.3}$ | $26.0_{\pm9.1}$ | $60.5_{\pm11.1}$ | $76.6_{\pm5.2}$ | $80.4_{\pm4.3}$ |
| | *First-n* | $0.7_{\pm0.4}$ | $11.7_{\pm4.3}$ | $50.3_{\pm9.1}$ | $81.2_{\pm5.3}$ | $84.8_{\pm5.0}$ |
| | Top-n | $\mathbf{8.3}_{\pm1.9}$ | $\mathbf{48.2}_{\pm6.4}$ | $\mathbf{86.4}_{\pm3.8}$ | $\mathbf{93.0}_{\pm2.6}$ | $\mathbf{94.1}_{\pm2.3}$ |

Table 4: Mean and 95% confidence interval over 5 runs on synthetic molecule-like data in 3d.

| | Train | Test | | Generation | | | Extrapolation | |
|---|---|---|---|---|---|---|---|---|
| | $\mathcal{W}_2$ distance | $\mathcal{W}_2$ distance | Valency loss | Incorrect valency | Diversity score | Valency loss | Incorrect valency | Diversity score |
| MLP | $\mathbf{0.47}_{\pm.07}$ | $\mathbf{0.42}_{\pm.03}$ | $0.73_{\pm.06}$ | $22.4_{\pm2.2}$ | $4.6_{\pm.1}$ | $1.06_{\pm.14}$ | $26.6_{\pm1.9}$ | $4.3_{\pm.2}$ |
| i.i.d. | $1.20_{\pm.01}$ | $0.81_{\pm.12}$ | $1.40_{\pm.07}$ | $36.8_{\pm20.9}$ | $5.0_{\pm.3}$ | $\mathbf{0.24}_{\pm.06}$ | $\mathbf{7.8}_{\pm0.5}$ | $\mathbf{4.9}_{\pm.2}$ |
| First-n | $\mathbf{0.43}_{\pm.08}$ | $0.50_{\pm.07}$ | $0.84_{\pm.06}$ | $24.6_{\pm2.1}$ | $5.1_{\pm.3}$ | $0.81_{\pm.19}$ | $22.5_{\pm3.0}$ | $\mathbf{4.6}_{\pm.3}$ |
| Top-n | $0.58_{\pm.05}$ | $\mathbf{0.44}_{\pm.03}$ | $\mathbf{0.37}_{\pm.12}$ | $\mathbf{13.9}_{\pm3.7}$ | $4.8_{\pm.2}$ | $0.80_{\pm.10}$ | $17.0_{\pm1.8}$ | $4.5_{\pm.2}$ |

The model is trained on 10 DSPN iterations and evaluated on 30, which is the setting that gave the best results in (Zhang et al., 2019).

Results are presented in Tables 2 and 3. For bounding box prediction (which is a simpler task), independent sampling and Top-n outperform MLP and First-n creation. As the metrics are computed for test data (which is not the case in SetMNIST), these results suggest that MLP and First-n creation may overfit the training images. On the full scene prediction, Top-n outperforms all other methods.

## 5.3 SYNTHETIC DATASET

As previous datasets do not measure generation quality, we further benchmark the different set generators on a synthetic dataset for which the quality of the generated sets can be assessed. This dataset is a simplified model of molecules in 3D, and retains some of its characteristics: i) atoms are never too close to each other ii) there is a bond between two atoms if and only if they are closer than a given distance (that depends on the atom types) iii) if formal charges are forbidden, each atom has a predefined valency. The generation procedure is described in Appendix D.2.

The goal is to reconstruct the atom positions and generate new realistic sets. During training, we measure the Wasserstein ($\mathcal{W}_2$) distance between the input and reconstructed sets. At generation time, we compute the $\mathcal{W}_2$ distance between the distribution of valencies in the dataset and in the generated set, the proportion of generated atoms with valency 0 or more than 4, as well as a diversity score to ensure that a method does not always generate the same set. We also measure the same metrics in an extrapolation setting where sets have on average 10 more points.

We train a VAE with different creation methods on this dataset. Details about the model and the loss function can be found in Appendix D.2, as well as an ablation study on the number of points in the reference set. The results are shown in Table 4. We observe that the independent sampling generator generalizes well but reconstructs the training sets very poorly, which reflects the fact that the model is hard to train due to the stochastic i.i.d. generation. They tend to generate points that are too far apart, an issue which disappears when generating more points. On the contrary, MLP and First-n

Table 5: Molecular graph generation on QM9. Baseline results are from the original authors. Our architecture provides an effective approach to one-shot molecule generation. Apart from independent sampling creation, the different set creation methods seem to be equivalent in this setting.

| Method | Generator | Valid (%) | Unique and valid |
|---|---|---|---|
| Graph VAE (Simonovsky & Komodakis, 2018) | MLP | 55.7 | 42.3 |
| Graph VAE + RL (Kwon et al., 2019) | MLP | 94.5 | 32.4 |
| MolGAN (De Cao & Kipf, 2018) | MLP | 98.0 | 2.3 |
| GTVAE (Mitton et al., 2021) | MLP | 74.6 | 16.8 |
| Set2GraphVAE (ours) | MLP | $60.5_{\pm 2.2}$ | $\mathbf{55.4}_{\pm 2.3}$ |
| | i.i.d. sampling | $34.9_{\pm 15.2}$ | $29.9_{\pm 10.0}$ |
| | First-n | $59.9_{\pm 2.7}$ | $\mathbf{56.2}_{\pm 2.7}$ |
| | Top-n | $59.9_{\pm 1.4}$ | $\mathbf{56.2}_{\pm 1.1}$ |

overfit the training data at the expense of generation quality. Finally, Top-n is able to generate points that have the right valency and generates the best new samples.

### 5.4 MOLECULAR GRAPH GENERATION

Finally, we evaluate Top-n on a graph generation task. We train a graph VAE (detailed in Appendix D.3) on QM9 molecules and check its ability to generate a wide range of valid molecules. As a generation metric, we simply report the validity (i.e., the proportion of molecules that satisfy a set of basic chemical rules) and uniqueness (the proportion non-isomorphic graphs among valid ones) of the generated molecules. We do not report novelty, as QM9 is an enumeration of all possible molecules up to 9 heavy atoms that satisfy a predefined set of constraints (Ruddigkeit et al., 2012; Ramakrishnan et al., 2014): in this setting, generating novel molecules is therefore not an indicator of good performance, but rather a sign that the distribution of the training data has not been properly captured. Note that most recently proposed methods proposed on this task use a non-learned validity correction code to generate almost 100% of valid molecules, which obfuscates the real performance of the learned model. We therefore only compare to works that do not correct validity. We also note that recursive methods such as Li et al. (2018) can often generate higher rates of valid molecules because they can easily check at each step that the added edge will not break valency constraints. They have therefore an unfair advantage over one-shot models which do not incorporate these checks.

While MolGAN and GraphVAE+RL both suffer from mode collapse, our method is able to generate a higher rate of valid and unique molecules. We observe that the independent sampling method is not able to obtain a good train loss (Appendix E), which reflects in the poor generation performance as well. The three other set creation methods seem to perform similarly. Our interpretation is that almost all molecules in QM9 have the same size (9 heavy atoms, since hydrogens are not represented), which makes it less important to properly handle varying graph sizes as in Top-n.

## 6 CONCLUSION

In this work, we strengthened the theoretical foundations of one-shot set and graph generation. We showed that, contrary to common belief, exchangeability is not a required property in GANs and VAE. We then proposed Top-n, a non-exchangeable model for one-shot set and graph creation which is able to select the most relevant points for each set. Our method can be incorporated in any GAN or VAE architecture and replace favorably other set creation methods.

We finally note that most of our experiments feature only small sets, which is in line with existing literature (Meldgaard et al., 2021; Satorras et al., 2021; Mitton et al., 2021). This observation leads to a simple question: can set and graph generation methods avoid the curse of dimensionality? For MLP-based generators, the answer is probably no: as they learn vectors of size $n_{\max}d$, standard universal approximation results are likely to yield a sample complexity that is exponential in $n_{\max}$. Whether other set generators can overcome this problem is an important question that conditions the success of one-shot models for larger sets and graphs.

**Ethics statement** Although our method can be used to generate any type of sets or graphs, it was primarily built and benchmarked with molecule generation applications in mind. While learning based methods will probably not replace expert knowledge in the short term, they can already be used to generate *hit* molecules, i.e., small and simple molecules that have promising activity on a predefined target. These molecules can then be refined by computational chemists in order to optimize activity and develop new effective drugs. We are therefore convinced that research on set and graph generation can have a direct positive impact on society.

**Reproducibility statement** Code is included in the supplementary materials.

ACKNOWLEDGMENTS

Clément Vignac would like to thank the Swiss Data Science Center for supporting him through the PhD fellowship program (grant P18-11).

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

## A  PROOF OF LEMMA 1

**Generative adversarial networks**  Given a set generator $f$, a discriminator function $d$, and $\boldsymbol{X}_1, ..., \boldsymbol{X}_m$ a training dataset, the standard loss function for GANs is formulated as

$$l(f, d, \boldsymbol{X}_1, ..., \boldsymbol{X}_m) = \frac{1}{m} \sum_{i=1}^{m} \log(d(\boldsymbol{X}_i))] + \mathbb{E}_{\mathbb{Z}}[\log(1 - d(f(\boldsymbol{z}))).$$

In order to obtain $l(f, d, \boldsymbol{X}_1, ..., \boldsymbol{X}_m) = l(f, d, \pi_1.\boldsymbol{X}_1, ..., \pi_m.\boldsymbol{X}_m)$ for every choice of $\pi_i$, it is therefore sufficient to choose a permutation invariant discriminator.

**Autoencoder based models**  We assume that the autoencoder is made of a permutation invariant encoder *enc* and an arbitrary decoder $f$. For any set size $n$, set $\boldsymbol{X} \in \mathbb{R}^{n \times d}$ and permutation $\pi \in \mathbb{S}_n$ we have

$$
\begin{aligned}
l(\pi.\boldsymbol{X}, \hat{\boldsymbol{X}}) = l(\boldsymbol{X}, \hat{\boldsymbol{X}}) &\implies l(\pi.\boldsymbol{X}, f(enc(\boldsymbol{X})) = l(\boldsymbol{X}, f(enc(\boldsymbol{X})) \\
&\implies l(\pi.\boldsymbol{X}, f(enc(\pi.\boldsymbol{X})) = l(\boldsymbol{X}, f(enc(\boldsymbol{X})) \quad (enc \text{ is invariant}) \\
&\implies \nabla_\theta \, l(\pi.\boldsymbol{X}, f(enc(\pi.\boldsymbol{X}))) = \nabla_\theta \, l(\boldsymbol{X}, f(enc(\boldsymbol{X})))
\end{aligned}
$$

## B  LOSS FUNCTIONS FOR SETS

Chamfer's loss and the Wasserstein 2 distance are defined as

$$d_{\text{Cham}} = \sum_{1 \leq i \leq n} \min_{j \leq n'} \|\boldsymbol{x}_i - \boldsymbol{x}'_j\|_2^2 + \sum_{1 \leq j \leq n'} \min_{i \leq n} \|\boldsymbol{x}_i - \boldsymbol{x}'_j\|_2^2$$

$$d_{\mathcal{W}_2} = \inf_{u \in \{\Gamma(\boldsymbol{X}, \boldsymbol{X}')\}} \sum_{\substack{1 \leq i \leq n \\ 1 \leq j \leq n'}} u(\boldsymbol{x}_i, \boldsymbol{x}'_j) \, \|\boldsymbol{x}_i - \boldsymbol{x}'_j\|_2^2$$

where $\Gamma$ is the set of couplings (i.e., bistochastic matrices) between $\boldsymbol{X}$ and $\boldsymbol{X}'$. Both loss functions solve a matching problem over the space of permutations, which makes them invariant to permutations of one argument, as required by Lemma 1. Chamfer's loss runs in quadratic time, while the standard implementation of Wasserstein distance runs in $O(n^3)$ if both sets have the same size. However, efficient algorithms exist for approximating the Wasserstein distance, and the computations can be parallelized on GPUs (Cuturi, 2013; Feydy et al., 2019).

Note however that the equation $l(\pi.\boldsymbol{X}, \hat{\boldsymbol{X}}) = l(\boldsymbol{X}, \hat{\boldsymbol{X}})$ is may be difficult to satisfy in other settings: matching graphs up to permutations, or sets up to the $SE(3)$ group, leads to difficult problems for which no polynomial time algorithm is known (Mémoli, 2007). In these settings, the design of VAE is harder and other architectures may be better suited.

## C  PROOF OF PROPOSITION 2

*Proof.* We first give the proof for First-n creation:

**First-n**  Given a set $\boldsymbol{Y} = \{\boldsymbol{y}_1, ..., \boldsymbol{y}_n\}$ of points in $\mathbb{R}^d$, we propose to sort the points $\boldsymbol{y}_i$ by alphanumeric sort (sort the values using the first feature, then sort points that have the same first features along the second feature, etc.). We denote the resulting matrix by $\boldsymbol{Y}'$. We choose as a latent vector $\boldsymbol{z} = \textit{flatten}(\boldsymbol{Y}') \in \mathbb{R}^{nd}$. It is the vector that contains the representation of $\boldsymbol{y}'_1$ in the first $d$ features, $\boldsymbol{y}'_2$ is the next $d$ features... By construction, $\boldsymbol{z}$ is a permutation invariant representation of the set $\boldsymbol{Y}$ (this reflects the fact that there are canonical ordering for sets, which is not the case for general graphs).

We choose as a reference set the canonical basis in $\mathbb{R}^n$ ($\boldsymbol{R} = \boldsymbol{I}_n$). After the latent vector is appended to the representation of each point, we have $\boldsymbol{r}_i = (\boldsymbol{e}_i, \boldsymbol{z})$. We are now looking for a function $f$ that allows to approximate the set $\boldsymbol{Y}$, i.e., that satisfies $\forall i \leq n, f(\boldsymbol{e}_i, \boldsymbol{z}) = \boldsymbol{z}[i\ d : (i+1)d]$. We can choose $f(\boldsymbol{e}_i, \boldsymbol{z}) = \boldsymbol{e}_i^T \boldsymbol{W}_1 \boldsymbol{W}_2 \boldsymbol{z}$, where

$$\boldsymbol{W}_1 = \begin{bmatrix} 1 & 1 & 1 & 0 & 0 & 0 & & 0 & 0 & 0 \\ 0 & 0 & 0 & 1 & 1 & 1 & \dots & 0 & 0 & 0 \\ 0 & 0 & 0 & 0 & 0 & 0 & & 1 & 1 & 1 \end{bmatrix} \in \mathbb{R}^{n \times nd} \quad \text{and} \quad \boldsymbol{W}_2 = \begin{bmatrix} 1 & 0 & 0 \\ 0 & 1 & 0 \\ 0 & 0 & 1 \\ & \dots & \\ 1 & 0 & 0 \\ 0 & 1 & 0 \\ 0 & 0 & 1 \end{bmatrix} \in \mathbb{R}^{nd \times d}$$

.

This function is continuous, and its output is $\boldsymbol{Y}' = reshape_{n \times d}(\boldsymbol{z})$. $\boldsymbol{Y}'$ is equal to $\boldsymbol{Y}$ up to a permutation of the rows, so that $||\boldsymbol{Y} - \boldsymbol{Y}'||_{\mathcal{W}_2} = 0$. If the entries of $\boldsymbol{z}$ are bounded, we can use standard approximation results for continuous functions over a compact space (Cybenko, 1989) to conclude that it be uniformly approximated by a 2-layer MLP.

**Top-n** Consider a Top-n network with $n$ reference points such that:

- The angles of the reference points are 2d vectors such that $\phi_i = (\cos(\frac{i}{n}\frac{\pi}{4}), \sin(\frac{i}{n}\frac{\pi}{4}))$.
- The representations are $r_i = e_i / \cos(\frac{i}{n_0}\frac{\pi}{4})$, where $(e_i)$ is the canonical basis in $\mathbb{R}^n$.
- The MLP of equation 1 (that predicts an angle from the latent vector) always outputs $(1, 0)$.

Then this Top-n creation module is equivalent to the First-n module build previously: for any set, it selects the first $n$ points and returns $\boldsymbol{X}^0[i] = \boldsymbol{e}_i$. The same MLP that is built for First-n can therefore be used for this network. □

# D DETAILS ABOUT THE EXPERIMENAL SETTING

## D.1 SET MNIST AND CLEVR

Since we use existing code for this task, we refer to the respective papers (Zhang et al., 2019) and (Kosiorek et al., 2020) for details on the model used and the loss function. The code used for TSPN is not the original code (which is not available) but a reimplementation (not by one of the authors of the present paper). The reader will notice that our results for DSPN are approximately 3 times worse than in the original paper. The reason is that we fixed what we believe to be a bug in the implementation of Chamfer's loss in DSPN: a mean over channels was used instead of a sum, which explains this difference of a factor 3. We also note that the results for TSPN are around 3 times worse than the original paper. One possible reason could be that the authors of TSPN used the code of DSPN and had a similar bug.

For Top-n generation, we set the number of points in the reference set to twice the cardinality $n$ of the generated sets. We also experimented with $n_0 = n$ which resulted in better performance for DSPN (with a Chamfer loss of $6.14 \pm 0.56$ e-5), but not for TSPN ($16.07 \pm 0.47$). We observed that reducing the learning rate improves results for all methods: TSPN was therefore trained for 100 epochs with a learning rate of 5e-4, and DSPN with a learning rate of 1e-4 for 200 epochs. No other hyper-parameter was tuned.

## D.2 SYNTHETIC SET GENERATION

**Dataset generation procedure** Each set is created via rejection sampling: points are drawn iteratively from a uniform distribution within a bounding box in $\mathbb{R}^3$. The first point is always accepted, and the next ones are accepted only if they satisfy a predefined set of constraints: i) they are not closer to any other point than a given threshold *min-distance*. ii) they are connected to the rest of the set, i.e., have at least one neighbor than a distance *neighbor-distance* iii) they do not have too many neighbors. Our dataset is made of 2000 sets that have between 2 and 35 points, with 9 points per set on average. It is a simplification of real molecules in several aspects: there are only single

Table 6: Train reconstruction error and valency loss in the generated sets over 5 runs for a modified version of our dataset, where the cardinality varies less across sets. We observe a tradeoff between reconstruction performance and generalization.

| Reference points | 11 | 13 | 15 | 20 | 30 | 50 |
|---|---|---|---|---|---|---|
| $\mathcal{W}_2$ train loss | **0.75**±.04 | 0.78±.03 | 0.79±.04 | 0.87±.05 | 0.93±.04 | 1.03±.06 |
| Valency loss (e-1) | 2.8±.7 | 2.2±0.7 | 2.1±.9 | 2.4±1.2 | **1.6**±.2 | 2.8±.8 |

bonds, angles between bonds are not constrained and the atom types are only defined by the valency, which reflects the fact that atoms with the same valency tend to play a similar role and be more interchangeable.

**Model**   The set encoder is made of a 2-layer MLP, 3 transformer layers followed by a PNA global pooling layer (that computes the sum, mean, max and standard deviation over each channel) (Corso et al., 2020) and a 2-layer MLP. The decoder is made of a set generator followed by a linear layer, 3 transformer layers and a 2-layer MLP. We use residual connections when possible and batch normalization between each layer. The reference set contains 35 points.

We experimented with the two ways to sample the number of points presented in Section 2, but we found the results to be quite similar. We therefore opted sampling the number of points from the data distribution, which is the simplest method.

**Loss function**   We use a standard variational autoencoder loss with a Wasserstein reconstruction term and two additional regularizers. Given an input set $\boldsymbol{X}$ and its reconstruction $\hat{\boldsymbol{X}}$, the loss can be written:

$$L(\boldsymbol{X}, \hat{\boldsymbol{X}}) = d_{W_2}(\boldsymbol{X}, \hat{\boldsymbol{X}}) + \lambda_1 \, KL(p(\boldsymbol{z} \mid \boldsymbol{X}), \mathcal{N}(0, \boldsymbol{I}_l)) + \lambda_2 reg_2 \, (\hat{\boldsymbol{X}}) + \lambda_3 \, reg_3(\hat{\boldsymbol{X}})$$

where

$$reg_2(\boldsymbol{X}) = \sum_{1 \leq i < j \leq n} (d_0 - ||\boldsymbol{x}_i - \boldsymbol{x}_j||_2)_+ \quad \text{with} \quad d_0 = 1$$

prevents atoms from being generated too close to each other. $reg_3(\boldsymbol{X})$ penalizes atoms that have either no neighbor, or a too large valency. It is computed in the following way:

- for each point $i$, compute $s^i = \text{sort}((d_{ij})_{j \leq n, \, j \neq i})$. This vector contains the sorted distances between $i$ and all other points. Points that are at distance less than $d_1 = $ *neighbor-distance* from $i$ are considered as its neighbours.
- Compute $l_1(i) = (s_0^i - d_1)_+$. This term penalizes atoms that have no neighbour.
- Compute $l_2(i) = \sum_{j=max\text{-}valency}^{n-1} (d_1 - s_j^i)_+$. This term penalizes atoms that have too many neighbors.
- $reg_3(X)$ is defined as $\sum_{1 \leq i \leq n} l_1(i) + l_2(i)$.

**Training details**   In order to train the model efficiently, mini-batches have to be used. This may not be easy when dealing with sets and graphs, since they do not have all the same shape. To circumvent this issue, we reorganise the training data in order to ensure that all sets inside a mini-batch have the same size. At generation time, this method cannot be applied, so we simply generate sets one by one.

The optimizer is Adam with its default parameters. We use a learning rate of $2e^{-4}$ and a scheduler that halves it when reconstruction performance does not improve significantly after 750 epochs. Experimentally, we found the learning rate decay to be important to achieve good reconstruction.

We also run a study with different reference set sizes. For this purpose, we slightly modify our dataset so that each set has only up to 11 points (still with 9 points on average). The reason is that there is more flexibility in the choice of the reference size if the maximal size is not too large. By training a Top-n network with several reference set sizes, we obtain the results of Table 6.

### D.3  MOLECULAR GRAPH GENERATION ON QM9

**Model**  Our encoder is a graph neural network is made of 3 message-passing layers followed by a PNA global pooling layer and a final MLP. For the decoder, we use a set creation method followed by transformer layers. The resulting representation is then processed by i) a Set2Graph layer (Serviansky et al., 2020) followed by two MLPs to generate edge probabilities and edge features a MLP which generates node features ii) a MLP that takes as input the set representation and the valencies predicted for each atom, and returns an atom type.

**Graph matching and loss function**  As explained in Appendix B, the loss function of the variational autoencoder should solve a graph matching problem, which is hard in general. Instead of using a proper graph matching method, we propose to use the atom types to perform an imperfect but much cheaper alignment between the target and the predicted molecules.

For both molecules, we compute a score for each atom $i$ defined as:

$$s(i) = 10^5 \textit{atom-type}(i) + 10^4 \textit{num-edges}(i) + \sum_{j \in \mathcal{N}(i)} \textit{edge-type}(i, j) * \textit{atom-type}(j)$$

This score cannot differentiate between all atoms in each molecule, but it reduces drastically the number of permutations that can represent the same input. It is motivated by the fact that empirically, we observe that our method quickly learns to reconstruct the molecular formula very well. Once they are all computed, we use these scores to sort the atoms reorder the adjacency matrix and the atom and edge types.

Our model learns to predict a probabilistic model for the atom types, edge presence and edge types. For this purpose, we use standard cross entropy loss between the predicted probabilities and the ground truth. However, these metrics can be hard to optimize because of the imperfect graph matching algorithm. We therefore regularize these metrics with several other measures at the graph level, that do not depend on matching:

- The mean squared error between the real atomic formula and the predicted one.
- The mean squared error between the average number of edges per atom in the input and predicted molecule.
- The mean squared error between the distribution of edge types in the real and predicted molecule.

Finally, we add a matching dependent term, which is the mean squared error between the valencies of the input and the target molecule.

**Training details**  The model is trained over 600 epochs with a batch size of 512 and a learning rate of $2e^{-3}$. It is halved after 100 epochs when the loss does not improve anymore. The optimizer is Adam with default parameters. The reference set has 12 points. When using more points, we obtained overall similar results, but with a larger variance.

# E  TRAINING CURVES

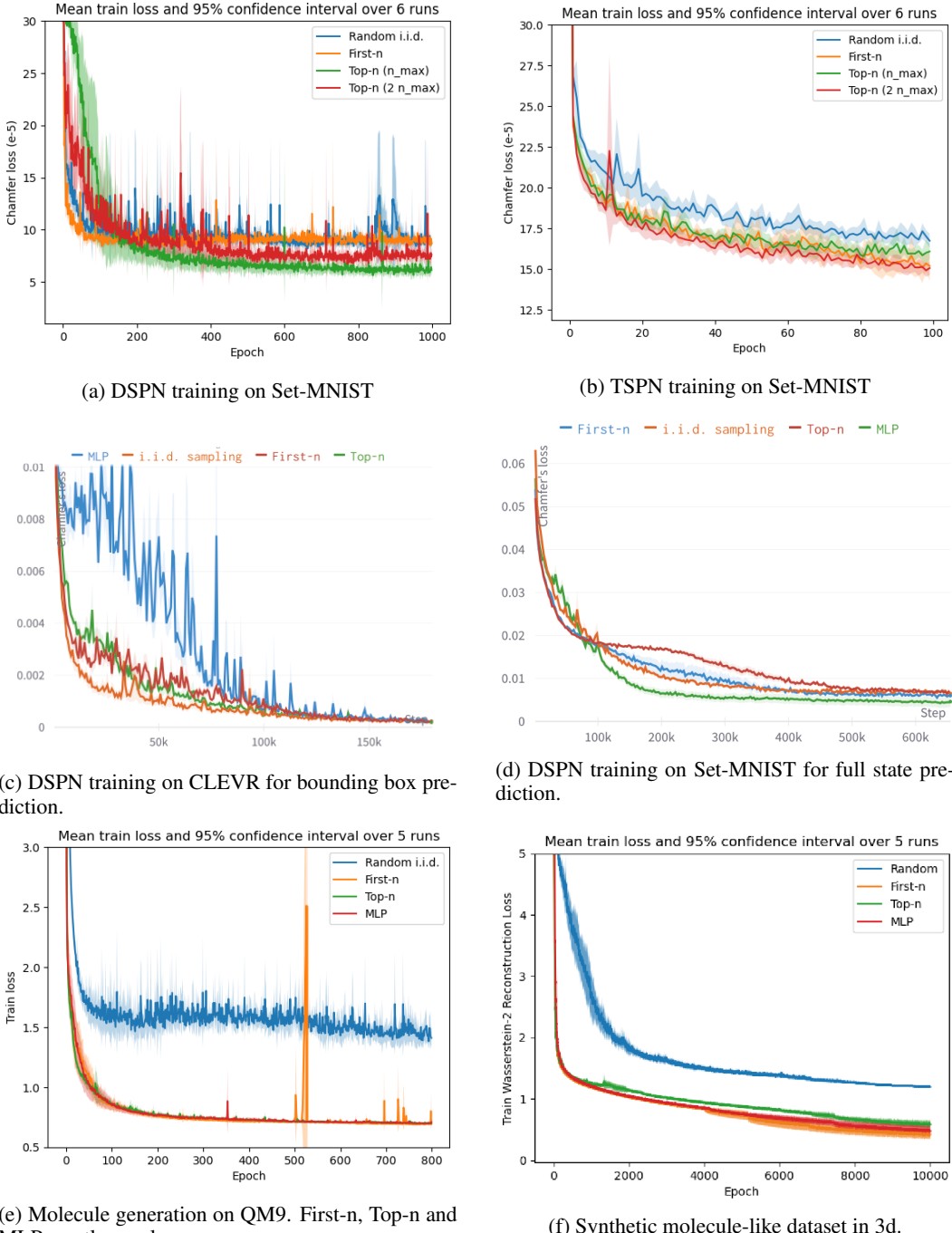

(a) DSPN training on Set-MNIST

(b) TSPN training on Set-MNIST

(c) DSPN training on CLEVR for bounding box prediction.

(d) DSPN training on Set-MNIST for full state prediction.

(e) Molecule generation on QM9. First-n, Top-n and MLP mostly overlap.

(f) Synthetic molecule-like dataset in 3d.

Figure 4: Training curves for all models. We observe that random i.i.d. generation is in general harder to train than the other models, while the differences between the other methods are smaller.

