# OpenReview forum: "Top-N: Equivariant Set and Graph Generation without Exchangeability"
_ICLR.cc/2022/Conference — ICLR 2022 Poster_

### Official Review · Reviewer_tgJk · 2021-10-29

**Correctness:** 2
**Technical Novelty And Significance:** 2
**Empirical Novelty And Significance:** 2
**Recommendation:** 5
**Confidence:** 4

**Main Review:**

**strengths**
* set creation and graph generation is a relevant topic to the DL community, with interesting applications to drug discovery, as the authors mention.
* new ways to handle permutation invariance and equivariance in graphs and sets is also a timely topic.


**weaknesses**

* No evidence is provided for claims. In several parts of the paper the authors claim that i.i.d. set creation methods are harder to train due to additional stochasticity. For example in section 5.1: the authors claim i.i.d. creation methods need more epochs to be trained than first-n and top-n. No evidence is provided here.

* I have several issues with the definition of equivariance (definition 1) and lemma 1:
  - Definition 1: If an algorithm is equivariant if dynamics do not depend on group elements, what would it mean for an algorithm to be invariant?
  - 3rd point in lemma 1 is not new (as indicated by authors) and not an extension of existing ways of viewing equivariance.
  - The first point is something that is already proposed in MolGAN by De Cao and Kipf (2018). In that paper the discriminator and reward function is invariant to node permutations. What is new here? In section 2.1 The authors state that MolGAN by De Cao & Kipf 2018 uses an MLP strategy that doesn’t take into account equivariance/invariance with respect to permutations. It is true that the generator does not have any equivariance or permutation invariance, but the discriminator and reward function use permutation invariant architectures, so according to the definition of this paper, it is just as "equivariant" as the proposed top-n method.

*  In general the paper contains many unclear statements that I doubt are all correct. Examples:
    - Second paragraph on page 2: "architectures based on i.i.d. creation are more flexible and generalise better than set creation alternatives based on multi-layer perceptrons (MLP)." I don't understand why, please explain and also give references to work in which this was shown.
    - First paragraph section 2. "f(z) should have the same law as D". What does that mean? f(z) represents a decoding distribution p(X|z) right? Whereas D is the marginal data distribution p(X).
    - Paragraph above def 1: "this definition can unfortunately not be used for generative modeling: since the input is a latent vector and not a set, it cannot be permuted". This statement is very unclear. Just like rows of matrices can be permuted, elements of a vector can also be permuted.
    - Last bullet point in page 2 "the true value of n is used during training" What is the true value here?
    - Bottom of page 3:  "both components make it harder to reconstruct the training data". Why is it harder to reconstruct the training data if you have two stochastic components. You are doing probabilistic inference, not pure pointwise reconstruction. Since it is a generative model and not a pure reconstruction model, I don't really see the problem here.
    - Bottom of page 3: "It is revealing to observe that most methods use a very low-dimensional normal distribution (sometimes in 1d), which is not enough to provide interesting features."  this is a vague claim that is not backed up by any references or otherwise convincing reasoning.
    - Equations for loss at the bottom of page 4 are inconsistent. In first eq. the first argument is the input X, and the second is f_theta (unclear what f_theta is exactly). In the second and third eq. in which the loss appears in the text, the first entry is f(X) (a prediction?) and the second argument is Y (a label?).
    - How does pi act on X in second paragraph of page 5? I don’t understand the argumentation here.
    - Page 5: The couplings in d_W2 are not defined. The text above this equation implies the couplings must be permutation matrices, but couplings could also be doubly stochastic matrices.
    - Proof of proposition 1 (appendix C, section header should refer to proposition 1, not 2): the authors state "as first-n creation is a special case of top-n generation (…) it is sufficient to prove the property for first-n creation". Proving something for a special case does not mean it’s valid for the general case.
    - it is unclear how the mechanism for making sorting differentiable works in eq. 3. It’s not sufficiently clear from the paper, which doesn't contain much more information than a reference to Gao & Ji.
    - What is X^{(0)} in eq on page 6?
    - Just below proposition 1: I don’t understand how f(X,z) in the eq. on page 7 translates to a normalizing flow with a fixed-width MLP. A normalizing flow takes in as input a random variable, and spits out a random variable of the same size. This doesn’t necessarily mean that the MLP's in the neural  network need to have a fixed hidden dim. See RealNVP where the translation network can have arbitrary hidden dims.
    - The loss function for a VAE in Section D2:  the kl divergence should be between an approximate posterior and a prior. What is p(z|W)? What is W? the posterior should be conditioned on X, not W. reg 3 is not defined and the reader is referred to the code. Please provide a definition in the paper.
    - At the end of section 5 the authors state that one-shot methods such as molgan and graph vae sample edges and nodes from bernoulli and multinomial distributions and they state that these models make an assumption that all nodes and edges are independent. This is not correct. Even if  a VAE decoder treats nodes conditionally independent, (conditioned on the latent variable z), that doesn’t mean that the marginal over nodes is factorized (after integration of the latent z).

* Comments on experiments:
    - Why is there no comparison against GG-GAN by  Krawczuk et al 2021 for SetMNIST?
    - On SetMNIST, the results for TSPN are not the same as reported in the paper (as mentioned by the authors). They are significantly worse (15.45 in this paper vs 5.42 in original paper). It's even worse than the DSPN baselines. Please explain why this reimplementation has such a big difference? Have you reached out to the authors? The non-anonymized link of the reimplementation is given in the paper. Is this a reimplementation done by the authors of this submission or is it made by someone else?
   - SetMNIST: top-n is not that much better than first-n for TSPN, large overlapping confidence intervals.
   - SetMNIST: "better performance" is attributed to the compositional nature of setmnist, but argumentation for this is unclear.
   - A new synthetic molecule-like dataset is created in section 5.2. What is the added benefit of this dataset over QM9? I don't see any new insights here.
   - Section 5.3 states that novelty is not reported because QM9 contains an enumeration of all possible molecules up to 9 heavy atoms. According to this reasoning, a novelty score would always be zero if you compared against the entire QM9 dataset. However, several baselines such as  MolGAN don't report 0% novelty in their manuscripts.  It makes more sense to compare against the training set, which means that even if QM9 would contain all possible molecules up to 9 heavy atoms, the test set would contain molecules not seen by the model and this would still be a valid metric.

* The introduction does contain a discussion of one-shot vs sequential graph generation, but no references are included.


**Minor comments/questions**
* DSPN is not defined in point 2 of lemma 1.
* Please be more specific on how W1 and W2 are concatenated below the last eq in page 6.
* Should the first sentence in the second paragraph of 5.1 refer to table 1 instead of figure 1?
* Please include details of TSPN and DSPN in the appendix as you use their implementations.

# post rebuttal
I have increased my score to a 5 based on the rebuttal. See the discussion below for more details on the reasoning.

**Summary Of The Paper:**

This submission discusses probabilistic models that generate sets and graphs conditioned on latent vector representations. The paper discusses the following different approaches to generate sets:
* i.i.d sampling of set element representations, concatenated with the set's vector representation followed by (possibly equivariant) networks.
* First-n generation, using a learnable reference set of a maximum number of nodes/set elements, and concatenating the latent set representation to each element of this reference set, and then picking the first n element of this reference set as node representations.
* MLP-based generation of node representations that don't take into account invariance of the generator under node permutation.

The authors claim that i.i.d sampling has two problems: the additional stochasticity makes it harder to train, and if two points are sampled too close together the resulting node representations will be similar (dubbed the collision problem).
Furthermore, MLP-based generators don’t explicitly take into account permutation symmetries, can't generalize to arbitrary number of nodes and first-n generation prioritizes learning of the first elements of the reference list as they will be used more often than the last elements of this reference list.

The authors propose a new definition of equivariance with respect to permutations: instead of talking about equivariance of functions, they propose to define equivariance for a learning algorithm, by stating that
"a learning algorithm is equivariant to the action of a group if the training dynamics do not depend on the group elements that are used to represent the training data".

They furthermore adapt first-n to the method top-n with differentiable sorting of the representations of the elements in the reference set based on cosine similarity with a vector that depends on the latent vector representation.

The following claims are made:
*  top-n is easier to train than i.i.d generation because it doesn't involve an extra sampling step
* top-n  captures complex dependencies in data better.

The proposed method is benchmarked on set and graph generation tasks : SetMNIST, synthetic molecule-like 3D structures, the QM9 dataset.


**Summary Of The Review:**

Although the topic of the paper is very relevant, there are too many significant weaknesses listed in the main review for me to be able to recommend acceptance at this point. I welcome clarifications by the authors.

---

> ### Author Response · Authors · 2021-11-11
> **Answer to reviewer tgJk (1/n)**
>
> We would like to thank the reviewer for this very detailed feedback, that we believe will help improve the paper quality a lot. Given the number of points raised by the reviewer, we will answer the different questions progressively. We hope that it will encourage discussion if some points remain unclear. Finally, we will upload a new version of the paper at the end of the rebuttal period, so that you can make sure that your concerns have been adressed.
>
>  1. *In several parts of the paper the authors claim that i.i.d. set creation methods are harder to train  due to additional stochasticity. For example in section 5.1: the authors claim i.i.d. creation methods need more epochs to be trained than  first-n and top-n.*
>
> We thank you for drawing our attention on the fact that this important aspect is not properly documented. This phenomenon has been observed by the authors of GG-GAN (Section 3.2, Figure 2): https://pasteboard.co/TgywPBkIrrmq.png. It also matches our experience: to show it, we will include training curves for all experiments in an appendix. They can currently be found at: https://pasteboard.co/CnNXfkWmD27S.png, https://pasteboard.co/I7dXxJtDkfvg.png, https://pasteboard.co/MZec2JfhPMQS.png, https://pasteboard.co/0C5AgbMEuMEp.png.
>
>  2. *Definition 1: If an algorithm is equivariant if dynamics do not  depend on group elements, what would it mean for an algorithm to be  invariant?*
>
> To answer this question, it may be useful to explain the  mathematical definition of an equivariant function in the general case. Using the notations of [Kondor18], a neural network $\phi$ is equivariant if there exists a representation $T_g$ that acts on the input space and a representation $T_{g}'$ that acts on the output space, such that for any input $f$, $\phi (T_g f) = T_{g}' ~ \phi(f)$. When $T_{g} = I_d$ (the identity matrix), we call this property invariance -- but invariance is actually a special case of the general definition of equivariance.
>
> [Kondor18]: http://proceedings.mlr.press/v80/kondor18a/kondor18a.pdf
>
> Because equivariance includes invariance, we chose to define an *equivariant learning algorithm* as well. We note that in the conditions of lemma 1, it does not matter what group representations are used. For any of them, the training dynamics will not depend on the group element if the conditions are met.
>
> 3. *3rd point in lemma 1 is not new (as indicated by authors) and not an extension of existing ways of viewing equivariance.  The first point is something that is already proposed in MolGAN by  De Cao and Kipf (2018). In that paper the discriminator and reward  function is invariant to node permutations. What is new here?*
>
> The goal of lemma 1 is not to provide new conditions that previous algorithms did not satisfy, but rather to give a formal justification to common practice. Virtually all GAN architectures use a permutation invariant discriminator, but we are the first to provide a theoretical framework that explains why this is indeed the right thing to do.
>
>  4. *In section 2.1 The authors state that MolGAN by De Cao & Kipf 2018 uses an MLP strategy that doesn’t take into account equivariance/invariance with  respect to permutations. It is true that the generator does not have any equivariance or permutation invariance, but the discriminator and  reward function use permutation invariant architectures, so according to the definition of this paper, it is just as "equivariant" as the  proposed top-n method.*
>
> Absolutely. The point we want to make is that MLPs do not yield exchangeable distributions, which was believed to be important for good performance. Yet, as proved in lemma 1, a learning algorithm based on a MLP decoder can still be equivariant. This explains why MLPs can work well in practice, as shown in section 5.3.
>
>  5.  In general the paper contains many unclear statements that I doubt are all correct. Examples: Second paragraph on page 2: "architectures based on i.i.d. creation  are more flexible and generalise better than set creation alternatives  based on multi-layer perceptrons (MLP)." I don't understand why, please  explain and also give references to work in which this was shown.
>
> This statement indeed needs to be made more precise. We will replace it by "Contrary to MLPs, architectures based on i.i.d. creation can generate any number of points, and decouple the number of generated points from the number of trainable parameters in the model."
>
>  6. "f(z) should have the same law as D".  What does that mean? f(z) represents a decoding distribution p(X|z)  right?
>
> Informally, If $z$ is a random variable drawn from $p_Z(z)$, $f(z)$ is a random variable with law $p_Z(z) p(X|z)$.
>
> Formally, the law of the $f(z)$ is called the pushforward measure $f_*(p_z)$ of $p_z$ by $f$. It is defined as:
> for any B in the $\sigma$-algebra of $X$, $f_*(p_z)(B) = p_z(f^{-1})(B)$.
> The model should be such that $f_*(p_z) \approx \mathcal D$.
> We will clarify this aspect in the paper.

---

> ### Author Response · Authors · 2021-11-11
> **Answer to reviewer tgJk (2/n)**
>
>  7. *Paragraph above def 1: "this definition can unfortunately not be  used for generative modeling: since the input is a latent vector and not a set, it cannot be permuted". This statement is very unclear. Just  like rows of matrices can be permuted, elements of a vector can also be  permuted.*
>
> If the latent representation was a set $X \in \mathbb R^{n \times l}$, the probabilistic decoder would map a latent set to a new set, which falls into the standard framework of equivariant representation learning.
>
> In contrast, the entries of a vector in $\mathbb R^{l}$ do not represent individual node, and in this case the latent space is *invariant* to the action of the permutation group. The probabilistic decoder therefore maps an invariant latent space (of vectors) to an equivariant space (of sets). This case cannot be handled with the standard definition of equivariant or invariant functions.
>
>  8. *Last bullet point in page 2 "the true value of n is used during training" What is the true value here?*
>
> The decoder tries to predict the number of points $n$ in the training set from the latent vector $z$: $\hat n = MLP(z)$. To train this MLP, an auxiliary loss  is used. However, during training, the decoder generates $n$ points and not $\hat n$ points. The predicted value $\hat n$ is only used to sample new sets.
>
> Quote from TSPN paper: "While training TSPN, we use the ground-truth set-cardinality to instantiate the initial set, and we separately train the MLP by minimizing categorical cross-entropy with the ground- truth set sizes. The cardinality-MLP is used only at test-time."
>
>  9. *Bottom of page 3:  "both components make it harder to reconstruct  the training data". Why is it harder to reconstruct the training data if you have two stochastic components. You are doing probabilistic  inference, not pure pointwise reconstruction.  Since it is a generative model and not a pure reconstruction model, I don't really see the  problem here*
>
> The reviewer is raising an interesting question on the discrepancy between the theory and practice of VAEs. To answer it, we will use the notations of [KW14], where the encoder $g$ is parametrized by $\phi$ and the decoder $f$ by $\theta$.
>
> [KW14]: https://arxiv.org/pdf/1312.6114.pdf
>
> The loss function derived from the ELBO writes: $-L(\theta, \phi, x^i) = \mathbb E_{q_\phi (z | x^i)} \log p_\theta(x^i | z) - D_{KL}(q_\phi(z|x^i)||p_\theta(z))$.
>
> In the first term, the amount of stochasticity in the decoder should not be important, as all that matters is the likelihood of the resulting distribution. In practice however, this term is approximated by a reconstruction loss between the input and the decoder output. This reconstruction loss is unfortunately only a proxy to -$ \mathbb E_{q_\phi (z | x^i)} \log p_\theta(x^i | z)$. The likelihood term and the reconstruction loss share the same optimal solution (the one that allows to reconstruct the data), but they are not equivalent in general.
>
> As a result, a stochastic decoder would not be an issue if we could directly maximize $\log p_\theta(x^i | z)$, but the practice of VAEs is unfortunately closer to pure point wise reconstruction.
>
> 10. *"It is revealing to observe that most methods use a very low-dimensional normal distribution (sometimes in 1d), which is  not enough to provide interesting features."*
>
> The normal distributions that we found in the litterature have the following dimension:
>
> NF-EGNN (Satorras et al.) and eq-NODE (Kohler et al. 20) : 2d and 3d.
>
> Pointflow (Yang et al. 19): 3d
>
> Graph EBM (Liu et al. 21): 6d
>
> TSPN (Kosiorec et al.), GAST (Stelzner et al.): unknown.
>
> Experimentally, we also found on the set generation task that increasing the dimensionality of points drawn i.i.d. did not improve performance.
>
> We note however that in most papers, the input dimensionality needs to match the dimensions of the output space (for normalizing flows or energy based models especially). It could therefore be that the only reason why low-dimensional distributions are used is that current benchmarks deal with generation of sets in a 2d or 3d spaces.
> As for the statement "sometimes in 1d", it comes from a misunderstanding of equation 7 of TSPN, in which the dimension is in fact not specified (and code is not available).
> We will remove this sentence altogether, as it is not key to the understanding of the paper.
>
>  11. *Equations for loss at the bottom of page 4 are inconsistent. In  first eq. the first argument is the input X, and the second is f_theta  (unclear what f_theta is exactly). In the second and third eq. in which  the loss appears in the text, the first entry is f(X) (a prediction?)  and the second argument is Y (a label?).*
>
> We understand that these notations are confusing. We will split the cases of unsupervised learning and supervised learning, and write the respective loss functions $l_u(X; f)$ and $l_s(f(X), Y)$.

---

> ### Author Response · Authors · 2021-11-12
> **Answer to reviewer tgJk (3/n)**
>
>  12. *How does pi act on X in second paragraph of page 5?.*
>
> $X$ is a $n \times d$ matrix representing a set. $\pi$ acts on it by permuting its rows.
>
>  13. *Page 5: The couplings in d_W2 are not defined. The text above this  equation implies the couplings must be permutation matrices, but  couplings could also be doubly stochastic matrices.*
>
> The set of possibles couplings is indeed the set of bistochastic matrices. The potential solutions can still be seen as a generalized matching problem, where mass from one point may be split across several target points. In fact, if the two sets have the same cardinality, the optimal coupling actually does actually not split mass [PC18] (Proposition 2.1, page 15).
>
>    [PC18]: https://arxiv.org/pdf/1803.00567.pdf
>
> 14. *Proof of proposition 1 (appendix C, section header should refer to  proposition 1, not 2): the authors state "as first-n creation is a  special case of top-n generation (…) it is sufficient to prove the  property for first-n creation". Proving something for a special case  does not mean it’s valid for the general case*
>
> The statement is in the form "it exists", which is why it is sufficient to prove it for First-n. The same two-layer MLP that is built for First-n can be used for Top-n.
>
> 15. *It is unclear how the mechanism for making sorting differentiable  works in eq. 3. It’s not sufficiently clear from the paper, which  doesn't contain much more information than a reference to Gao & Ji*.
>
> Thank you for telling us that this important point is not clear enough. Equation 3 itself is not differentiable, so gradients do not flow through the sorting operation: $\frac{\partial R[s]}{\partial \Theta} = 0$. What makes it possible to train the MLP of equation (1) and the angle vectors $\Theta$ is the modulation of $R[s]$ with $\tilde c = \text{softmax}(c[s])$ in equation (5). Using the chain rule we have for example:
> $\frac{d X}{d \Theta} =\frac{\partial X}{\partial R[s]} \frac{d R[s]}{d \Theta} + \frac{\partial X}{\partial \tilde c} \frac{d \tilde c}{d \Theta}  = 0 + \frac{\partial X}{\partial \tilde c} \frac{d \tilde c}{d \Theta}$ which is not 0 in general. Equation (5) can therefore be seen as a reparametrization trick that indirectly allows to obtain gradients through the sorting operation.
>
> 16. *What is X^{(0)} in eq on page 6?*
>
> It is a typo. $X^0$ should simply be $X$.
>
> 17. *Just below proposition 1: I don’t understand how f(X,z) in the eq.  on page 7 translates to a normalizing flow with a fixed-width MLP. A  normalizing flow takes in as input a random variable, and spits out a  random variable of the same size. This doesn’t necessarily mean that the MLP's in the neural  network need to have a fixed hidden dim. See  RealNVP where the translation network can have arbitrary hidden dims.*
>
> To simplify the problem, we can first consider the problem of building a normalizing flow $f$ based on First-n: a reference set $X_{ref} \in \mathbb R^{n \times c}$ is given, the latent $z \in \mathbb R^{l \times 1}$ is the only stochastic component, and the initial set can be written $cat(X_\text{ref}, 1_n z^T)$.
>
> Given an arbitrary set $X \in \mathbb R^{n \times d}$, $f$ should be such that $f^{-1}(X)$ has the form  $cat(X_\text{ref}, 1_n z_0^T)$ for some vector $z_0$. If it is not the case, the associated likelihood is 0 and the model cannot be trained or evaluated.
>
> Proposition 1 states that if the two first layers of $f$ are a MLP, $f^{-1}(X)$ can be arbitrarily close to some set of the form $cat(X_\text{ref}, 1_n z_0^T)$ . However, it is not clear how to build a normalizing flow that satisfies this property. For example, one could think of a RealNVP that is applied independently on each row. The result would probably be such that $f^{-1}(X) = cat(X_\text{ref}, X')$ with no guarantee that $X'$ can be written as $1_n z_0^T$. We therefore believe that First-n is not suited to normalizing flows.
>
> Additional issues appear when using Top-n, since it is built using a hard selection process (the value of the reference points that are not selected is not used at all). Changes to the model are therefore required to make it invertible.

---

> ### Author Response · Authors · 2021-11-17
> **Answer to reviewer tgJk (4/n)**
>
>   18. *The loss function for a VAE in Section D2:  the kl divergence should be between an approximate posterior and a prior. What is p(z|W)? What  is W? the posterior should be conditioned on X, not W. reg 3 is not  defined and the reader is referred to the code. Please provide a  definition in the paper.*
>
> It is a typo, $W$ should be $X$.
>
> $reg_3$ is computed in the following way:
>
>   - for each point $i$, compute $s^i=\text{sort}((d_{ij})_{j \leq n,~ j\neq i})$. This vector contains the sorted distances between $i$ and all other points. Points that are at distance less than $d_1$=*neighbor-distance* from $i$ are considered as its neighbours.
>    - Compute $l_1(i) = (s^i_0 - d_1)_+ $. This term penalizes atoms that have no neighbour.
>    - Compute $l_2(i) = \sum_{j=max-valency}^{n-1} (d_1 - s^i_j)_+$. This term penalizes atoms that have too many neighbors.
>    - $reg_3(X)$ is defined as $\sum_{1 \leq i \leq n} l_1(i) + l_2(i)$.
>
>   19. *At the end of section 5 the authors state that one-shot methods such as molgan and graph vae sample edges and nodes from bernoulli and  multinomial distributions and they state that these models make an  assumption that all nodes and edges are independent. This is not  correct. Even if  a VAE decoder treats nodes conditionally independent,  (conditioned on the latent variable z), that doesn’t mean that the  marginal over nodes is factorized (after integration of the latent z).*
>
> We thank the reviewer for spotting this error. It came from an experimental observation about the Graph Transformer model we built, which would generate several molecules when trained to overfit only one molecule. This observation was incorrectly generalized to all one-shot models, while it was simply due to a problem in our implementation.
>
> Based on your comment, we fixed our model, which should improve the performance of all implemented methods. We will share the results and update the paper accordiningly.
>
>   20. *Comments on experiments: Why is there no comparison against GG-GAN by  Krawczuk et al 2021 for SetMNIST?*
>
> GG-Gan is a graph generation method, and is therefore not benchmarked on SetMNIST. For future work, we will consider either applying Top-n to their experimental protocol, or reimplementing GG-GAN on QM9.
>
>   21. *On SetMNIST, the results for TSPN are not the same as reported in  the paper (as mentioned by the authors). They are significantly worse  (15.45 in this paper vs 5.42 in original paper). It's even worse than  the DSPN baselines. Please explain why this reimplementation has such a  big difference? Have you reached out to the authors? The non-anonymized  link of the reimplementation is given in the paper. Is this a  reimplementation done by the authors of this submission or is it made by someone else?*
>
> As the source code is not provided in the TSPN paper, we used an implementation by someone else. We checked the code but did not find any mistake that could explain this performance gap. However, we noticed that not all numerical values are described in the paper. For example, the dimension of the points drawn i.i.d. is unknown. The reason why we still chose to use this code is that we don't aim to achieve state of the art results on this benchmark, but rather to show that Top-N can be use as a replacement to other methods and improve performance.
>
>   22. *SetMNIST: "better performance" is attributed to the compositional nature of setmnist, but argumentation for this is unclear.*
>
> The motivations for Top-n (with respect to First-n) are to give the network the ability to extrapolate to larger sizes than seen during training, to remove the bias in First-n due to the fact that some points are selected more often than other, and to create a mechanism that can learn to select the points that are most relevant for a given set.
>
> In SetMNIST, all sets have the same cardinality (360 points), so that the two first motivations cannot explain why Top-n outperforms First-n. It therefore seems that Top-n indeed manages to select the right points for each set.
>
> Out understanding is that the location of the points can roughly be guessed from the class label of the set, and that Top-n manages to use this information to provide varied reference sets that are suited to the task.
>
>   23. A new synthetic molecule-like dataset is created in section 5.2.  What is the added benefit of this dataset over QM9?*
>
> The main motivation for the synthetic dataset is to create a setting where there are not bond types, so that a set can be predicted rather than a graph. It removes the need for graph matching, which makes training easier.

---

> ### Author Response · Authors · 2021-11-17
> **Answer to reviewer tgJk (5/n, n=5)**
>
> 24. *Section 5.3 states that novelty is not reported because QM9 contains an enumeration of all possible molecules up to 9 heavy atoms. According to this reasoning, a novelty score would always be zero if you compared against the entire QM9 dataset. However, several baselines such as   MolGAN don't report 0% novelty in their manuscripts.*
>
> We will add references supporting our claim that QM9 is an enumeration:
>
> - From [RDBR12]: "we report the enumeration of organic molecules up to 17 atoms of C, N, O, S, and halogens, forming the chemical universe database GDB-17 containing 166.4 billion organic molecules."
> - From [QM9]: "These molecules correspond to the subset of all 133,885 species with up to nine heavy atoms (CONF) out of the GDB-17 chemical universe"
>
> [RDBR12]: https://pubs.acs.org/doi/10.1021/ci300415d
> [QM9]: http://quantum-machine.org/datasets/
>
> Our understanding is that novelty is not 0 in practice because several output smiles may correspond to the same stable molecule. For example, if there is a positive formal charge on one atom, a negative one on another atom, we may believe that we have a novel molecule while the corresponding molecule with 0 formal charges is present in the dataset. The molecules should first be "standardized" before measuring novelty.
>
> 25. *It makes more  sense to compare against the training set, which means that even if QM9  would contain all possible molecules up to 9 heavy atoms, the test set  would contain molecules not seen by the model and this would still be a  valid metric.*
>
> This metric seems to be an elegant way to measure generation quality. It is unfortunately not computed in the baselines that we consider, but it would be a useful complement to the standard metrics.
>
>
> 26. *The introduction does contain a discussion of one-shot vs sequential graph generation, but no references are included.*
>
> Thank you for this observation, we will add them.

---

> > ### Comment · Reviewer_tgJk · 2021-11-19
> > **reply 1/2 to rebuttal**
> >
> > Thank you for your extensive rebuttal. Below I comment on your reply:
> >
> > 1. Thank you for providing additional evidence for this claim. Have you tuned the learning rate separately for all methods in the plots that you show?
> > 2. Thank you for reiterating the standard definition of equivariance and invariance, I'm familiar with it. Let me rephrase my question: in the standard definition of equivariant functions, invariant functions are a special case, by setting T_g = I_d. What specific detail do I need to restrict in your definition of an equivariant learning algorithm to come to the special case of an invariant learning algorithm?
> > 3. I don’t think you're actually giving a formal justification to a common practice. From my point of view you're reiterating a common practice and giving it a new name, which by itself is not that valuable.
> > 4. Similar to my point 3, lemma 1 does not give an explanation for why MolGAN or MLP-based generators work well despite not yielding exchangeable distributions. It was clear from previous work that using a permutation invariant discriminator is good and plays a crucial role, even if the generator is itself not equivariant. Your explanation for why this works is just to give a name to it. I don't see the additional insight.
> > 5. Good clarification.
> > 6. OK.
> > 7. I think I understand what you mean, but your answer still contains statements that are confusing. You are saying the latent space is invariant and the output of the decoder is an equivariant space. Shouldn't we be speaking of equivariant functions and not equivariant spaces.
> > 8. OK
> > 9. I strongly disagree here.  The stochasticity in the decoder does play a role in the first term in the ELBO, both in theory and in practice. For example, if the decoder distribution is Gaussian with a learnable mean and a fixed variance, this leads to an expectation with respect to z, over a scaled mean squared error between a mean that is conditioned on z and the data input. The fixed variance comes into play through the scaling of the reconstruction loss, which influences how it compares to the KL term.  Such a reconstruction loss is not an approximation of the first term in the elbo, it is exact. The only approximation comes in the form of a Monte Carlo estimate of the expectation, but that is unbiased (using the reparameterization trick). A VAE is certainly not optimized to perform a point wise reconstruction of datapoints, not in theory and not in practice. The second part of the ELBO (the KL term) ensures this.
> > 10. My problem was with "which is not enough to provide interesting features", which is a vague statement. But if you take it out that works for me.
> > 11. OK
> > 12. OK
> > 13. OK. Please define the set of allowed couplings in the paper.
> > 14. I don't know what "The statement is in the form "it exists"" means. If you say the same MLP can be used for First-n as for Top-n, then please just explain that. I repeat that proving something for a special case doesn't make it hold for the more general case automatically, you have to explain why that is the case here.
> > 15. Is eq 3 by any chance an arg sort, not a sort, given that the output of eq 3 are indices? I understand that in eq 5 there are still other ways in which X depends in a differentiable manner on the parameters theta, namely through $\tilde c$. However, there is no gradient flowing through the sorting operation. Eq. 5 just has other parts that _are_ differentiable. So I disagree that eq 5 is a reparameterization trick to obtain gradients through the sorting operation. There is extensive literature on actual differentiable sorting algorithms, e.g. https://arxiv.org/abs/2002.08871, which is quite different from what you mean with "obtaining gradients through a sorting operation".
> > 16. OK
> > 17. I don’t understand your reply. If your base random variable is cat(X_ref, 1 z_0^T) you need to define a distribution over this random variable, otherwise you can't perform the change of variables as is usually done in normalizing flows. In this case X_ref is not a random variable but a constant, so it has no density (or it has a delta dirac distribution if you wish). So I completely agree that proposition 1 can't be used with normalizing flows, but I don't see why it has something to do with a fixed width of an MLP.

---

> > > ### Comment · Reviewer_tgJk · 2021-11-19
> > > **reply 2/2 to rebuttal**
> > >
> > > 18. OK
> > > 19. Ok
> > > 20. Ok
> > > 21. I understand it is difficult if source code of a baseline is not provided. And I agree that you don't always have to show state of the art performance. Showing that your method can be combined with others and improves performance is certainly useful. However, if the discrepancy between the reproduction and the originally reported numbers is as large as it is in the current case, then that casts doubts on the validity of the implementation (or of the original work). It therefore also casts doubt on the reliability of the result that using top-n improves performance here.
> > > 22. This is a much clearer explanation. However in the paper you say it has to do with compositionality. I don't understand what being able to learn to select points has to do with the concept of compositionality.
> > > 23. Ok.
> > > 24. I agree that several output smiles can correspond to the same stable molecule. However, I don't think QM9 is exhaustive in the sense that it contains all possible molecules with up to 9 heavy atoms that are possible in the world. It is true that QM9 is the subset of GDB-17 of all molecules up to 9 heavy atoms. But I don't think GDB-17 itself contains all molecules up to 17 heavy atoms that are chemically possible "in the universe".  An example statement from the paper in which GDB-17 was introduced even states this: 'On the other hand, GDB-17 represents a selective enumeration and therefore does not contain all molecules in the reference databases. Overall 57% of PubChem-17, 60% of ChEMBL-17, and 68% of DrugBank-17 are compatible with the GDB-17 enumeration rules. The molecules found in the reference databases but not considered for GDB-17 contain nonenumerated features such as certain types of halogens (e.g., aliphatic halogens) or sulfurs (thiols, thioethers, thioureas), functional groups (e.g., acyclic acetals, hemiacetals, aminals, azides, aliphatic nitro groups), elements (P, Si, B, etc.), skeletons (nonaromatic C═C), or graphs (e.g., spiro-fused cyclopropanes) (Figure 4A).' This clearly states there are molecules that are not part of GDB-17 because they don't follow the rules used to generate molecules in this dataset.
> > > 25. OK
> > > 26. OK

---

> > > > ### Author Response · Authors · 2021-11-19
> > > > **Discussion (1/4)**
> > > >
> > > > Thank you for this answer, we hope to address all your concerns below:
> > > >
> > > > 1. *i.i.d. set creation is harder to train* --  Have you tuned the learning rate separately for all methods in the plots that you show?
> > > >
> > > > In preliminary experiments, we checked the largest learning rate that could be used for each method. We observed that First-n could in general be trained with a slightly higher learning rate than the other methods, but that the differences in the maximal learning rates were overall not very important (all the more that we use Adam and a learning rate scheduler). In the final experiments, we therefore chose to use the same learning rate for each method.
> > > >
> > > > 2. *definition equivariant learning algorithm* -- Thank you for reiterating the standard definition of equivariance and invariance, I'm familiar with it. Let me rephrase my question: in the standard definition of equivariant functions, invariant functions are a special case, by setting T_g = I_d. What specific detail do I need to restrict in your definition of an equivariant learning algorithm to come to the special case of an invariant learning algorithm?
> > > >
> > > > Sorry, we meant no disrespect by stating the standard definition of equivariance, we simply did it because it was not written in the paper. There is no definition of an invariant learning algorithm, because our definition does not depend on whether the group action is trivial or not -- we could however have called our definition "invariant learning algorithm" as well.
> > > >
> > > > We removed references to permutation invariance in this section in order to avoid this confusion.

---

> > > > > ### Author Response · Authors · 2021-11-19
> > > > > **Discussion (2/4)**
> > > > >
> > > > >   3. *added value of lemma 1* --  I don’t think you're actually giving a formal justification to a common practice. From my point of view you're reiterating a common practice and giving it a new name, which by itself is not that valuable.  4. Similar to my point 3, lemma 1 does not give an explanation for why MolGAN or MLP-based generators work well despite not yielding exchangeable distributions. It was clear from previous work that using a permutation invariant discriminator is good and plays a crucial role, even if the generator is itself not equivariant. Your explanation for why this works is just to give a name to it. I don't see the additional insight.
> > > > >
> > > > > Based on your review, we significantly revised this section in order to better highlight our contribution.
> > > > >
> > > > > In particular, we made our definition of equivariance more precise and gave it another name (even if it is fundamentally the same). We now define it for a pair $(F_\Theta, l)$ where $F_\Theta= \{f_\theta: \mathcal X \to \mathcal Y; ~\theta \in \Theta\}$ is a model class of functions from $\mathcal X$ to $\mathcal Y$ (for example a neural architecture parametrized by $\theta$), and $l$ a loss function:
> > > > >
> > > > > > Consider a model class $F_\theta \subset \mathcal Y ^ \mathcal X$, a group $G$ that acts on $\mathcal X$ and $\mathcal Y$ and a loss function $l$ defined on $\mathcal Y$. We say that the pair $(F_\Theta, l)$ is equivariant to the action of $G$ if the dynamics of $\theta \in \Theta$ trained with gradient descent on $l$ do not depend on the group elements that are used to represent the training data.
> > > > >
> > > > > The reasoning that led to this section is the following:
> > > > >
> > > > > - Common opinion is that generative models should use an invariant loss function and an exchangeable probabilistic decoder. To our knowledge, the only setting in which this opinion is well supported is for normalizing flow models. In particular, it is not clear why exchangeable methods should be better than methods that generates the same sets, but with a fixed permutation.
> > > > >
> > > > > - In discriminative models, most focus has been put on the design of equivariant functions. In practice, we notice that they are always used with invariant loss functions (i.e., functions that satisfy $\forall f \in F,~ \forall g \in G,~ \forall (X, Y) \in \mathcal X \times \mathcal Y, \quad l(g . f(X), g . Y) = l(f(X), Y)$). While it clearly seems like the right think to do, we are not aware of any explanation of why invariant loss functions should be used.
> > > > > - Our observation is the following: when combining an equivariant model and an invariant loss, the training dynamics become independent of the group elements used to represent the data.
> > > > > - We therefore define as $(F, l)$ equivariant the class of architectures which training dynamics do not depend on the group action.
> > > > > - We analyze what this definition corresponds to for generative models. We cannot obtain necessary conditions for $(F, l)$ equivariance, because all sorts of architecture might by chance be equivariant (in the same way that it is theoretically possible to build an equivariant function by stacking non-equivariant layers). However, very elementary proofs allow to obtain the sufficient conditions of Lemma 1. These conditions are more relaxed than for discriminative models, because they do not impose that the probabilistic decoder be equivariant.
> > > > > - We observe that (virtually) all existing architectures achieve these sufficient conditions. We held this as a good sign that our definition of equivariance is actually relevant and important for good experimental performance.
> > > > > - For GANs and VAEs, these conditions do not require that the probabilistic decoder should be exchangeable. This is in line with the observation that exchangeable models do not really perform better in practice, and in particular that most graph generation methods are based on MLP creation.
> > > > >
> > > > > 7. *The standard definition of equivariance for probabilistic decoders* -- I think I understand what you mean, but your answer still contains statements that are confusing. You are saying the latent space is invariant and the output of the decoder is an equivariant space. Shouldn't we be speaking of equivariant functions and not equivariant spaces.
> > > > >
> > > > > We called  (maybe abusively) *invariant latent space* a space on which the action of G is trivial, and *equivariant* a space on which it is not. For more clarity, we changed this paragraph as follows:
> > > > >
> > > > > >  In discriminative models, symmetries are accounted for when a neural network $f$ is equivariant to the action of the permutation group, which writes $\pi . f(X) = f(\pi . X)$ . This definition can unfortunately not be used for generative modeling with a latent vector: imposing $\pi .f(z) = f(\pi. z) = f(z)$ would only allow for trivial solutions where all rows are equal, which is too restrictive.

---

> > > > > > ### Author Response · Authors · 2021-11-19
> > > > > > **Discussion (3/4)**
> > > > > >
> > > > > > 9.  *Training VAE is more difficult with a lot of stochasticity* --  I strongly disagree here. The stochasticity in the decoder does play a role in the first term in the ELBO, both in theory and in practice. The only approximation comes in the form of a Monte Carlo estimate of the expectation, but that is unbiased (using the reparameterization trick). A VAE is certainly not optimized to perform a point wise reconstruction of datapoints, not in theory and not in practice. The second part of the ELBO (the KL term) ensures this.
> > > > > >
> > > > > > While we are not experts in variational inference and the difficulty of training VAEs with random i.i.d. creation was mostly observed experimentally, our theoretical understanding is the following:
> > > > > >
> > > > > > Consider a training set $X$, and denote $p_{X^0}$  the distribution over the initial sets $X^0$ . For random i.i.d. creation, it is a multivariate Gaussian with a diagonal covariance matrix.
> > > > > >
> > > > > > With random i.i.d. creation, the probabilistic decoder is stochastic given $z$ , so that $\log p_\theta(X^i | z)$ cannot be computed in close form. To estimate the expectation, we instead write
> > > > > >
> > > > > > $\mathbb E_{q_\phi (z | x^i)} [\log p_\theta(x^i | z)] = \mathbb E_{X^0 \sim p_{X^0}} \big[\mathbb  E_{z \sim q_\phi (z | x^i)} [ \log p_\theta(x^i | z, X^0)] \big]$.
> > > > > >
> > > > > > The Monte Carlo estimator that we use computes $\log p_\theta(X | z, X^0)$. Because $X^0$ is independent of $z$, the variance of this estimator is larger than it would be if $X^0$ was deterministic, which is why it is harder to train in practice.
> > > > > >
> > > > > >   13.  Please define the set of allowed couplings in the paper.
> > > > > >
> > > > > > The paper has been modified accordingly.
> > > > > >
> > > > > >   14. *Proposition 1*-- I don't know what "The statement is in the form "it exists"" means. If you say the same MLP can be used for First-n as for Top-n, then please just explain that. I repeat that proving something for a special case doesn't make it hold for the more general case automatically, you have to explain why that is the case here.
> > > > > >
> > > > > > Yes, the same MLP can be used for First-n than for Top-n, sorry that it was not clear.
> > > > > >
> > > > > > To be more precise, consider a Top-n network with $n$ reference points such that:
> > > > > >
> > > > > > - The angles of the reference points are 2d vectors such that $\phi_i = (\cos(\frac{i}{n} \frac{\pi}{4}), \sin(\frac{i}{n} \frac{\pi}{4}))$.
> > > > > > - The representations are $r_i = e_i / \cos(\frac{i}{n_0} \frac{\pi}{4})$, where $(e_i)$ is the canonical basis in $\mathbb R^n$.
> > > > > > - The MLP of equation 1 (that predicts an angle from the latent vector) always outputs $(1, 0)$.
> > > > > >
> > > > > > Then this Top-n creation module is equivalent to the First-n module presented in the proof. The same MLP that is built for First-n can therefore be used for this network.
> > > > > >
> > > > > > 15.  *Top-n equations*-- Is eq 3 by any chance an arg sort, not a sort, given that the output of eq 3 are indices? I disagree that eq 5 is a reparameterization trick to obtain gradients through the sorting operation. There is extensive literature on actual differentiable sorting algorithms, e.g. https://arxiv.org/abs/2002.08871, which is quite different from what you mean with "obtaining gradients through a sorting operation".
> > > > > >
> > > > > > It is indeed an argsort. We agree that our method is very different from the differentiable sorting algorithms such as the one you refer to. We propose for example to replace this sentence by: "Eq (5). provides a path in the computational graph that does not go through the sorting operation, so that $d X^0/d \Phi$ is not always 0. "  Would this sentence be more clear?
> > > > > >
> > > > > > 17. *Top-n in normalizing flows*-- I don’t understand your reply. If your base random variable is cat(X_ref, 1 z_0^T) you need to define a distribution over this random variable, otherwise you can't perform the change of variables as is usually done in normalizing flows. In this case X_ref is not a random variable but a constant, so it has no density (or it has a delta dirac distribution if you wish). So I completely agree that proposition 1 can't be used with normalizing flows, but I don't see why it has something to do with a fixed width of an MLP.
> > > > > >
> > > > > > The issue is the following: given an arbitrary test set $X$, how to make sure that the likelihood of $f^{-1}(X)$ is not 0? If the set creation $g$ is the random i.i.d. method and $h$ is invertible, it is clear that $f=h\circ g$ will be surjective (as the composition of surjective functions).
> > > > > >
> > > > > > However, if $g$ is based on First-n, it is not surjective and it is not clear that $f$ can be surjective neither. What we show is that at least, with infinite width MLPs, any set $X$ can be *approximated* by a sequence $(f(z_i)_{i \in \mathbb N})$. Unfortunately, it is only an approximation result, and MLPs don't have infinite width in practice.
> > > > > >
> > > > > >   18.  *TSPN* -- It casts doubt on the reliability of the result that using top-n improves performance here.
> > > > > >
> > > > > > We understand your concerns.  Although this experiment is not the most reliable one, we believe that overall, our experiments still demonstrate the efficiency of Top-n.

---

> > > > > > > ### Author Response · Authors · 2021-11-19
> > > > > > > **Discussion (4/4)**
> > > > > > >
> > > > > > >   22. *Compositionality* -- This is a much clearer explanation. However in the paper you say it has to do with compositionality. I don't understand what being able to learn to select points has to do with the concept of compositionality.
> > > > > > >
> > > > > > > We removed all reference to compositionality in the paper. While our intuition is that more meaningful reference sets can be found when the data is organised in several clusters (such as MNIST digits), we agree that this claim is not supported by numbers.
> > > > > > >
> > > > > > > 23. *Novelty on QM9* -- I agree that several output smiles can correspond to the same stable molecule. However, I don't think QM9 is exhaustive in the sense that it contains all possible molecules with up to 9 heavy atoms that are possible in the world (+ references showing it).
> > > > > > >
> > > > > > > Thank you for this observation.
> > > > > > >
> > > > > > > We believe that what matters is that GDB-17 is *exhaustive within the set of constraints that was used to build it*. For example, if the dataset does not contain any aliphatic halogens, we probably don't want our algorithm to generate any. The ability of an algorithm to *not* generate substructures that have been removed from the dataset is actually used to benchmark drug discovery models (cf. [MOSES], Table 2, metric "Filters"). The rationale is that if a substructure is known to be toxic, then we don't want a drug discovery algorithm to generate it.
> > > > > > >
> > > > > > > [MOSES]:https://arxiv.org/pdf/1811.12823.pdf
> > > > > > >
> > > > > > > We therefore think that on QM9, novelty is not a good metric of generation quality. The situation would be different is the dataset was made of molecules that were experimentally found to be active on one target, for example. In the case of QM9, the measure that you proposed in your review (the coverage of a held-out test set) seems to be more pertinent.

---

> > > > > > > > ### Comment · Reviewer_tgJk · 2021-12-01
> > > > > > > > **response**
> > > > > > > >
> > > > > > > > Thank you to the authors for engaging in discussions. I appreciate that the authors have taken the time to run some additional experiments such as the ablation for the size of the reference set, and that they have provided clarifications for several of my questions and updated the manuscript.
> > > > > > > >
> > > > > > > > As I see it, the paper has two contributions:  1) defining a new perspective on equivariance and concluding that this definition shows that exchangeability of generative models is not necessary, and 2) proposing top-n.
> > > > > > > >
> > > > > > > > My concern about the new perspective on equivariance remains that I don't see the added value. Instead of the usual definition of equivariant functions, the definition of an equivariant learning algorithm is given, and sufficient conditions for equivariant learning algorithms for GANS and VAEs and Normalizing flows are given. However, as also stated by the authors, many existing architectures already satisfy these definitions:  previous GAN methods already use discriminators that are invariant to the symmetry under consideration (see molGAN), and VAE methods for sets already use Chamfer or matching-based losses. The authors then claim that the use of for instance these set-based losses wasn't motivated before and that the proposed definition serves as a good motivation. I disagree that it wasn't motivated properly before. It just so happens that previous papers didn't write their explanation in the form of a formal definition statement, but explained the motivation in words.
> > > > > > > >
> > > > > > > > Furthermore, the claim that "exchangeability is in fact unnecessary in VAEs and GANs" as long as the training dynamics is "equivariant" to permutations seems to me to be a too general statement. One can certainly come up with situations where you both want the training dynamics to be invariant to permutations, as well as have an exchangeable distribution. Example: if your goal is to model densities or wavefunctions of indistinguishable (bosonic) particles, you have to have a distribution that is symmetric under permutation of particle indices.
> > > > > > > >
> > > > > > > > The top-n method itself seems like a reasonable contribution, although the experimental validation has room for improvement. It is not clear to me what exactly changed in the code after the review period to improve the scores so much, and the discrepancy between the results for TSPN on setMNIST and that of the original paper remain curious.
> > > > > > > >
> > > > > > > > Finally, I think the rebuttal contains incorrect claims about the reconstruction term in a VAE for the i.i.d. generation method. It seems like the authors are saying that since the closed form of $p(x|z) = \int dX^0 p(x|z, X^0) p(X^0)$ is intractable, the reconstruction term becomes $\mathbb E_{q(z|x)} [\log p_{\theta}(x|z)] = E_{q(z|x)} \mathbb E_{p(X^0)} [\log p_{\theta}(x|z,X^0)]$. However, this  has to be an inequality instead of an equality, because the expectation over $X^0$ is taken out of the logarithm:
> > > > > > > > $\mathbb E_{q(z|x)} [\log p_{\theta}(x|z)] =  \mathbb E_{q(z|x)} [\log \mathbb E_{p(X^0)}[ p_{\theta}(x|z,X^0)] \geq \mathbb E_{q(z|x)} \mathbb E_{p(X^0)} [\log p_{\theta}(x|z,X^0)]$ due to Jensen's inequality.
> > > > > > > >
> > > > > > > > I will raise my score to a 5 instead of a 3 because of the improvements that have been made to the manuscript in terms of clarity. However, from my point of view, the above concerns are too big to increase the score further.

---

> ### Author Response · Authors · 2021-12-03
> **Post rebuttal**
>
> Dear reviewer, thank you for the score update.
>
> We now have an explanation for the discrepancy between the results of DSPN and TSPN.
>
> - In the original implentation of DSPN (that we are using), we spotted a mistake in Chamfer loss. A mean over the channels has been used instead of a sum in the computation of the Huber loss, *which results in a Chamfer loss 3 times smaller than it should be*.
>
> - The reimplentation of TSPN has the correct loss. We note that if the loss of DSPN is multiplied by 3, the reimplentation of TSPN becomes better than DSPN.
> - Since we don't have access to the code of TSPN, we cannot check whether it contains the same mistake. However, it seems likely, for several reasons: i) the paper seems to indicate that the code of TSPN is based on DSPN ii) The authors of TSPN replicate the results of the DSPN paper (which are 3 times too optimistic) iii) The reimplementation is almost exactly 3 times worse than TSPN (16.42 / 3 = 5.47 for the reimplementation versus 5.42 in the paper).
>
> We will update our results for DSPN (multiply the loss by 3), and explain the discrepancy with the original results.
>
>
> Regarding the examples that you mention (modeling indistinguishable particles), we agree that exchangeability is needed when modeling the density $p(X)$ of the matrix containing the state of the system.  However, it would also be possible (but less convenient) to model directly the density of the *set*:  p({$x_i$}$_{i \leq n}$). In this case exchangeability would not make sense, as it is defined for matrices. We believe that this phenonenon explains the difference between normalizing flows (where exchangeability is needed) and VAEs/GANs  (cf. footnote Page 5): normalizing flows manipulate ordered lists, while, for example, the input of a GAN discriminator can be considered as a unordered set.

---

### Official Review · Reviewer_uN3i · 2021-11-02

**Correctness:** 4
**Technical Novelty And Significance:** 3
**Empirical Novelty And Significance:** 2
**Recommendation:** 8
**Confidence:** 4

**Main Review:**

Strengths:
- The authors provide a good overview of prior work on generative models for sets, and in particular for the one-shot generation problem.
- The extension of the definition of equivariance from functions to learning algorithms in Sec. 3 is elegant and straightforward, and provides some theoretical basis for understanding why random iid generation in VAE and GANs does not outperform other methods such as MLPs.
- The proposed Top-n creation mechanism is presented clearly, and seems well motivated.  It appears to address problems with some existing set creation mechanisms, such as random iid generation and First-n creation, and empirically performs well.

Weaknesses:
- It would be helpful to include an experimental comparison with at least one recursive set generation method.  While the authors note in the Introduction that recursive methods are slow and introduce an ordering to the points in a set that does not exist in the data, in practice recursive methods may be competitive with one shot generation methods in terms of generative quality.
- The valency loss evaluation metric used in Table 2 is difficult to interpret.  Furthermore, while the training Wassertstein distance metric used in this table measures the model’s ability to reconstruct training sets, no similar evaluation metric is provided to directly evaluate the model’s ability to reconstruct sets at test time.  A more interpretable evaluation should be used here that more directly measures the ability of the model to construct realistic sets, such as measuring the Wasserstein distance at test time between generated sets and sets included in a held-out collection of test sets.



**Summary Of The Paper:**

This paper considers the problem of “one-shot” set/graph generation, which involves learning a probabilistic decoder that maps latent vectors to sets.  First, the authors extend the usual definition of equivariance for a function to a learning algorithm.  This definition is used to show that exchangeability is not useful in GANs and VAEs when used for set generation.  Next, the authors propose Top-n, which is a new set creation mechanism which learns to select the most relevant points from a trainable reference set, in a deterministic and non-exchangeable fashion.  Top-n can replace iid generation in a VAE or GAN.  Experimental results are provided for SetMNIST reconstruction and generative tasks for a synthetic molecule dataset for sets, and the QM9 chemical dataset for graphs, demonstrating that Top-n is competitive with or outperforms a number of existing generative approaches.

**Summary Of The Review:**

This paper has several reasonable contributions in the area of one-shot generative models for sets.  The extension of the notion of equivariance to learning algorithms is quite sensible and clearly presented, and the proposed Top-n set creation method is a solid contribution, providing significant improvements compared to competing approaches on several benchmark datasets and tasks.  There are a few issues with the experimental evaluation, as described above.

---

> ### Author Response · Authors · 2021-11-18
> **Answer to reviewer uN3i**
>
> We thank you for your reviewing work and useful suggestions. We adress below your concerns:
>
>   1. *It would be helpful to include an experimental comparison with at  least one recursive set generation method.  While the authors note in  the Introduction that recursive methods are slow and introduce an  ordering to the points in a set that does not exist in the data, in  practice recursive methods may be competitive with one shot generation  methods in terms of generative quality.*
>
> We agree that recursive methods can be competitive. In the case of molecule generation, they even outperform one-shot methods. We believe that the main reason is that most methods check valency constraints at each edge addition, which results in molecules that are valid most of the time.  Unfortunately, we found no reference of recursive generation method benchmarked no QM9. We will however add relevant references in the introduction and in the experimental section.
>
>   2. *The valency loss evaluation metric used in Table 2 is difficult to  interpret.  Furthermore, while the training Wassertstein distance metric used in this table measures the model’s ability to reconstruct training sets, no similar evaluation metric is provided to directly evaluate the model’s ability to reconstruct sets at test time.  A more interpretable evaluation should be used here that more directly measures the ability  of the model to construct realistic sets, such as measuring the  Wasserstein distance at test time between generated sets and sets included in a held-out collection of test sets.*
>
> We will add a test set reconstruction metric, which should nicely complement the W2 train metric. It is however still a reconstruction metric, that can check if a model overfits but does not measure generation quality. In particular, models that achieve good reconstruction do not necessarily generate good samples. This is for example the case of VAEs with a large latent dimension and a very small KL penalty, which do not really ensure that the latent distribution of the training data is Gaussian.
>
> Measuring generation quality is in general a challenge for all generative models. For example, measuring the Wasserstein distance between sampled sets and held-out sets might be useful when the data distribution lies in a small portion of the space, but it is not the case in general:  in our dataset there is no reason why the generated sets should be close to these particular test sets and not to other sets.
>
> Although we agree that our metric is not very easy to interpret, it has the advantage of measuring generation quality independently of a training set or a test set. This is the reason why we introduced this synthetic dataset.

---

> > ### Comment · Reviewer_uN3i · 2021-11-28
> > **Post-rebuttal feedback**
> >
> > I thank the authors for the revision of their paper and their extensive rebuttal comments.  The rebuttal comments and changes have helped to clarify the contributions of this paper, and the experimental results in Section 5.3 are stronger.  I'm satisfied with the authors' responses to the issues raised in my review, and it appears that they have addressed a number of questions and concerns from other reviewers.  In light of this, I will increase my score to an 8 (accept, good paper).  However, it would be very helpful to hear from the other reviews regarding whether or not their concerns have been addressed.

---

### Official Review · Reviewer_xa3Q · 2021-11-03

**Correctness:** 3
**Technical Novelty And Significance:** 3
**Empirical Novelty And Significance:** 3
**Recommendation:** 6
**Confidence:** 3

**Main Review:**

The paper makes significant contributions:

1. The author proposed a generalized definition of “equivariance”, which provided a new perspective for researchers to design generation model.

2. The proposed method Top-N in this paper integrates the advantages of previous models. The method has the ability to extrapolate to larger sets than those seen during training as well as the ability to train easily, which are not satisfied by the previous generative models simultaneously.

Despite the contributions mentioned above, I still have concerns below:

1. The experimental results in this paper using different models are compared on the basis of the generalized definition for equivariance. The results can prove the validity of the Top-N method to some extent, but can not justify exchangeability is unnecessary for the generative model. It is better to compare the traditional method which has the ability to generate exchangeable distributions with the one proposed in this paper. I think the author's point, “exchangeability is unnecessary for the generative model” ,can be better demonstrated in this case.

2. For set and graph generation tasks which should just satisfy the permutation invariance, it is relatively easy to design the model structure and choose the appropriate loss function in order to satisfying the generalized definition of equivariance. But for more complex generating tasks, for example, the task which should satisfy the symmetry of rotation, permutation, translation simultaneously, it is more difficult to design the full architecture. Therefore, I think the generalized definition is not practical for more complex generating tasks.

3. How to choose the size of the reference set? It is better to give several guidelines to balance training efficiency with model generalization ability, i.e. the ability to extrapolate to larger sets.

**Summary Of The Paper:**

In this paper, the author proposes that exchangeability is unnecessary for the generative model in the domain of set and graph generation. The definition of equivariance is generalized to learning algorithms, which is appropriate for generative modeling. Then, a method called Top-N which can be used in classical generative models, such as VAE, GANS, is proposed. In the author's argument, the proposed method has the ability to extrapolate to larger sets than those seen during training as well as the ability to train easily, which are not satisfied by the previous generative models.

**Summary Of The Review:**

Overall, I recommend accepting this paper but the authors are expected to address the concerns.

---

> ### Author Response · Authors · 2021-11-18
> **Answer to reviewer xa3Q**
>
> We would like to thank you for your work and useful suggestions. We adress below your concerns:
>
>   1. *The experimental  results in this paper using different models are compared on the basis of the generalized definition for equivariance. The results can prove  the validity of the Top-N method to some extent, but can not justify  exchangeability is unnecessary for the generative model. It is better to compare the traditional method which has the ability to generate  exchangeable distributions with the one proposed in this paper.*
>
>    We would like to stress two points:
>
>    - While exchangeability is not an important property for GANs and VAEs, it is not harmful neither. To understand why, we can consider a thought experiment where two models are trained on the same task. One is a function $f$ that generates sets with a fixed permutation, so that the output distribution is not exchangeable. The second model is $g= \textit{RANDOMIZE} \circ f$: it takes the output of $f$ and applies any permutation with equal probability. For any model $f$, the output distribution of $g$ is exchangeable. However, as long as a proper loss function for sets is used, $f$ and $g$ will be perfectly equivalent: they will generate the same sets and have the same training dynamics.
>    - In our experiments, we use 4 different set generation methods. The MLP, First-n and Top-n methods do not yield exchangeable distributions, but the random i.i.d. method does. As random i.i.d. generation is the most common method for set creation, it can be seen as the 'traditional exchangeable method' that is most often used. We will make this point clearer in the paper.
>
>   2. *For set and graph  generation tasks which should just satisfy the permutation invariance,  it is relatively easy to design the model structure and choose the  appropriate loss function in order to satisfying the generalized  definition of equivariance. But for more complex generating tasks, for example, the task which should satisfy the symmetry of rotation,  permutation, translation simultaneously, it is more difficult to design  the full architecture.*
>
>    The situation is very different for GANs and VAEs, as described in Lemma 1:
>
>    - When building a GAN architecture that is equivariant to several groups, it is sufficient to build a discriminator that is invariant to these groups. This is not a problem, as many algorithms have been proposed that are invariant to several groups simultaneously.
>    - When building a VAE architecture, the loss function has to match two sets up to the different groups. As explained in Section 3, this may indeed be difficult to do. For example, no polynomial time algorithm exists for matching sets up to translations, rotations and scalings. We however note that this problem is common to all VAE equivariant VAE architectures, and that it is not specific to Top-n.
>
>    In Section 3, we gave a definition of permutation equivariance for a learning algorithm, i.e. a procedure that takes as input a training set and a model class (=an architecture), and returns a predictor within this hypothesis class (=a trained model). While this definition is novel, it corresponds to what is commonly done in practice: for example, variational autoencoders for sets already use either Chamfer or Wasserstein loss, even if this choice was not properly motivated. We will elaborate on this point in the revised version.
>
>   3. *How to choose the  size of the reference set? It is better to give several guidelines to  balance training efficiency with model generalization ability, i.e. the  ability to extrapolate to larger sets.*
>
> The size of the reference set is indeed an hyperparameter that needs to be tuned. In general, it should be at least as large as the largest set size expected at test time. We observed that increasing the number of reference points reduces overfitting and leads to better generation metrics. However, if too many points are present in the reference set, each reference point will be update less often and will therefore not be trained efficiently. There is therefore a tradeoff. In practice, we found that, for datasets where the set size varies a lot, setting the reference size close to the larger set size is a safe choice.

---

> > ### Comment · Reviewer_xa3Q · 2021-12-05
> > **Thanks for reply**
> >
> > I appreciate the authors' reply. My concerns have been mainly addressed. I hope the authors can include the discussions here in their manuscript. My score keeps at 6.

---

### Official Review · Reviewer_tbNm · 2021-11-09

**Correctness:** 3
**Technical Novelty And Significance:** 3
**Empirical Novelty And Significance:** 2
**Recommendation:** 6
**Confidence:** 4

**Main Review:**

The paper is well written. Although I enjoyed reading, I feel that much of the effort is focused on describing set generation mechanism and exchangeability. For ease of readability and continuity, some of these parts can be moved to Appendix. For example, the whole section on exchangeability can be summed up in few lines. For VAE, matching loss will aid in avoiding exchangeability criteria and so on.

Feedback and clarification:

1. In Top-n mechanism the selection of reference points itself is non-differentiable i.e., Eq (3). How are gradients propagated ?
2. For Top-n algorithm, please elaborate on the dimension of W_i. In eq (6), if the dimension of z is $l$, I find it difficult to match the RHS and LHS dimension to be $n \times c$.
3. For reference points, how is 'angle' vector related to corresponding 'representation' vector ? Currently, I find it strange to have completely independent 'angle' vector for each of the reference point. Why can't one compute angle similar to that computed for latent vector ?
4. Does increase in cardinality of reference set improve performance ? Can you add ablation study on this ? At the same time the increase in reference set should lead to increase in stochasticity as each points are updated fewer times during training ? All I understand is each of this reference point is acting as a basis function. There has to be some theoretical lower bound connecting number of reference point with performance on each dataset.
5. Top-n model in Table 1 and 2 utilise twice the reference points as First-n model. Please include additional results with similar reference set as First-n model.
6. The equations for Chamfer loss and elsewhere are not numbered. I suggest that all the equations in the paper be numbered.
7. Can you clarify the explanation for normalising flow in proposition 1 ? All I understand is, during training, normalising flow model computes noise vector and its probability from underlying distribution. I believe Top-n algorithm is not reversible. And so it is not possible to map noise vector.
8. How do one select a reference set ?
9. Although the paper claims applicability of Top-n on GAN, I note that all the experimental results were rather evaluated for VAE architecture.
10. In order to properly benchmark against First-n mechanism, it would be more appropriate to compare models on the similar setting and tasks demonstrated in (Zhang et al., 2019a) (such as CLEVR data). Currently, I feel that final task of Molecular graph generation is not appropriate as,  a) the comparison is against the old baseline, b) many baseline generate all edges independently and c) similarity of this task to Synthetic dataset task.

**Summary Of The Paper:**

This work proposes a new deterministic set sampling mechanism, Top-n. Top-n learns to select the best 'n' points from a trainable reference set. Unlike the previous set sampling mechanisms such i.i.d. sampling, First-n and MLP projection, Top-n do not suffer from collision problem and can generate sets of various sizes (unseen during training). Top-n can be incorporated for one-shot sampling in VAE and GANs like generative models. Experimental results on standard benchmark for set and molecular graph generation, suggest improved performance in comparison to prior sampling mechanism.

**Summary Of The Review:**

Overall, to some extent the technical contribution is significant but experimental section should be further strengthen to justify the model and its applicability.

---

> ### Author Response · Authors · 2021-11-17
> **Answer to reviewer tbNm (1/2)**
>
> We thank the reviewer for the useful comments. In particular, we will take into account your general comment on the structure, and adapt the paper in order to have i) a shorter explanation of existing set creation methods ii) a clearer explanation of our notion of equivariance iii) more details about Top-n.
>
> We address your concerns below:
>
>   1. In Top-n mechanism the selection of reference points itself is non-differentiable i.e., Eq (3). How are gradients propagated ?
>
> This is an important point, and we will add additional explanations in the text. Equation 3 gives 0 gradients almost everywhere, so gradients do not flow through the sorting operation: $\frac{\partial R[s]}{\partial \Theta} = 0$. What makes it possible to train the MLP of equation (1) and the angle vectors $\Theta$ is the modulation of $R[s]$ with $\tilde c = \text{softmax}(c[s])$ in equation (5). Using the chain rule we have:
>
> $\frac{d X}{d \Theta} =\frac{\partial X}{\partial R[s]} \frac{d R[s]}{d \Theta} + \frac{\partial X}{\partial \tilde c} \frac{d \tilde c}{d \Theta}  = 0 + \frac{\partial X}{\partial \tilde c} \frac{d \tilde c}{d \Theta}$ which is not 0 in general (a similar equation can be written for the parameters of the MLP in Eq. (1)). Equation (5) can therefore be seen as a reparametrization trick that allows to obtain gradients through the sorting operation.
>
>   2. *For Top-n algorithm, please elaborate on the dimension of W_i. In eq (6), if the dimension of z is , I find it difficult to match the RHS and LHS dimension to be n x c.*
>
> There is indeed a mistake, thanks for noticing it. Equation 6 consists in applying at each point $i$ the operation  $x_i = x_i \odot z ~W_3 + z ~W_4$.  However, when considering the whole set, it should be written
>
> $X = X \odot 1_n ~ z^T ~W_3^T+ 1_n ~ z^T W_4^T$, where $1_n$ is a $n \times 1$ matrix of ones, z is seen as a $l  \times 1$ matrix, and $W_3, W_4$ have shape $l \times  c$.
>
>
>   3.  For reference points, how is 'angle' vector related to corresponding 'representation' vector ? Currently, I find it strange to have  completely independent 'angle' vector for each of the reference point.  Why can't one compute angle similar to that computed for latent vector ?
>
> The angle and the representation of each point are indeed parametrized independently. It is a difference with the Top-K pooling method of Gao & Ji that we should have stressed. In Top-K pooling, the selection of the points is based on the cosine between the latent vector and the representation. The drawback of this method is that it tends to select points that are similar. On the contrary, if the angles and the representation are independent, two points can have similar angles (i.e. be selected together) while having a very diverse representation.
>
> 4. Does increase in cardinality of reference set improve performance ?  Can you add ablation study on this ? At the same time the increase in  reference set should lead to increase in stochasticity as each points  are updated fewer times during training ? All I understand is each of  this reference point is acting as a basis function. There has to be some theoretical lower bound connecting number of reference point with  performance on each dataset.
>
> We encountered the tradeoff that you mention: in general, adding points to the reference set slows down training because each point is updated less often. We however noticed that models with more reference points can sometimes generalize better, despite poorer training metrics. It seems that models with few reference points are prone to overfitting, and adding more points can help.
>
> We will add an ablation study on this topic. Here are some preliminary results *on only one run*. We used a modified version of the synthetic dataset where the cardinality varies less across sets. There are still 9 points on average, but only up to 11 points.
>
> | Reference points               | 11    | 13    | 15    | 20    | 30    |
> |--------------------------------|-------|-------|-------|-------|-------|
> | Train Wasserstein distance     | 0.77  | 0.84  | 0.77  | 0.86  | 0.91  |
> | Valency distance at generation | 0.248 | 0.161 | 0.208 | 0.157 | 0.143 |
>
> Apart for the model with 15 points, we can see that models with more reference points do not reconstruct sets as well but generalize better.

---

> ### Author Response · Authors · 2021-11-17
> **Answer to reviewer tbNm (2/2)**
>
> 5. Top-n model in Table 1 and 2 utilise twice the reference points as  First-n model. Please include additional results with similar reference  set as First-n model.
>
>    We ran these experiments and obtained the following results:
>
>    | Method        |                  | TSPN           | DSPN          |
>    | ------------- | ---------------- | -------------- | ------------- |
>    | Set creation  | Reference points | Chamfer (e-5)  | Chamfer (e-5) |
>    | Random i.i.d. | --               | 16.42 +- 0.53  | 9.52 +- 0.41  |
>    | First-n       | n_max            | 15.45 +- 1.41  | 8.87 +-  0.18 |
>    | Top-n         | n_max            | 16.07 +- 0.47  | 6.14 +- 0.56  |
>    | Top-n         | 2 n_max          | 14.98 +- 0.59$ | 7.53 +- 0.57  |
>
> Reducing the number of reference points improved the results for DSPN but not for TSPN. It is important to note that the results for SetMNIST are only reconstruction metrics, and we don't know how these models would compare in terms of generation quality.
>
> 6. The equations for Chamfer loss and elsewhere are not numbered. I suggest that all the equations in the paper be numbered.
>
>    Thank you for the suggestion
>
> 7. Can you clarify the explanation for normalising flow in proposition 1 ? All I understand is, during training, normalising flow model computes noise vector and its probability from underlying distribution. I  believe Top-n algorithm is not reversible. And so it is not possible to  map noise vector.
>
> This explanation is correct: Top-n is built with a hard selection process, and is therefore not invertible. However, even a soft version of Top-n might not be suited to normalizing flows, since is it not even clear how to build a normalizing flow based on First-n:
>
>   - consider the problem of building a normalizing flow $f$ based on First-n: a reference set $X_{ref} \in \mathbb R^{n \times c}$ is given, the latent $z \in \mathbb R^{l \times 1}$ is the only stochastic component, and the initial set can be written $cat(X_\text{ref}, 1_n z^T)$.
>   - Given an arbitrary set $X \in \mathbb R^{n \times d}$, $f$ should be such that $f^{-1}(X)$ has the form  $cat(X_\text{ref}, 1_n z_0^T)$ for some vector $z_0$. If it is not the case, the associated likelihood is 0 and the model cannot be trained or evaluated.
>   - Proposition 1 states that if the two first layers of $f$ are a MLP, $f^{-1}(X)$ can be arbitrarily close to some set of the form $cat(X_\text{ref}, 1_n z_0^T)$ . However, it is not clear how to build a normalizing flow that satisfies this property. For example, one could think of using GraphNVP layers. The result would probably be such that $f^{-1}(X) = cat(X_\text{ref}, X')$ with no guarantee that $X'$ can be written as $1_n z_0^T$.
>
> [GraphNVP]: https://arxiv.org/pdf/1905.11600.pdf
>
>
>
> 8. How does one select a reference set ?
>
>    In our experiments, the points in the reference set are initialized randomly. We experimented with [Bridson's] algorithm to ensure that points are not too close to each other, but it did not improve performance: since the points are trainable, it seems the way they are initialized does not impact performance.
>
>    [Bridson's]: https://www.cs.ubc.ca/~rbridson/docs/bridson-siggraph07-poissondisk.pdf
>
> 8. Although the paper claims applicability of Top-n on GAN, I note that all the experimental results were rather evaluated for VAE  architecture.
>
>    We focused on VAEs because they are easier to train, but additional results on a GAN architecture would indeed be a good addition to our results.
>
> 9. In order to properly benchmark against First-n mechanism, it would  be more appropriate to compare models on the similar setting and tasks  demonstrated in (Zhang et al., 2019a) (such as CLEVR data). Currently, I feel that final task of Molecular graph generation is not appropriate  as,  a) the comparison is against the old baseline, b) many baseline  generate all edges independently and c) similarity of this task to  Synthetic dataset task.
>
>    We agree that results on CLEVER would be interesting as well. However, we believe that the results on QM9 are still meaningful because:
>
>    - Contrary to the synthetic dataset task (which is a set generation task), QM9 is a graph generation task. We find it important to show that Top-n can efficiently generate graphs when used with a Set2Graph function.
>    - Most recent one-shot methods use post-hoc code to correct the molecules and achieve higher validity. For example, we experimented with an energy based model for which validity went from 100% to less than 10% when we disabled the molecule correction.This model is not included in our results, because we could not make it work well without the validity correction.
>    - Similarly, most recurrent models check at each step that added edges will not create valency violations. For these reasons, they have an unfair advantage over one-shot models which do not incorporate these checks.

---

> ### Author Response · Authors · 2021-12-06
> **Did our response address your concerns?**
>
> Dear reviewer tbNm,
>
> we would be grateful if you confirm whether our response and revision have addressed your concerns. In addition to improving experimental results and rewriting the section on equivariance, we followed your advice and reorganised the paper in order to put more focus on Top-n. Please let us know if any issue remain.
>
> Thank you

---

> ### Comment · Reviewer_tbNm · 2021-12-09
> **Increasing my score**
>
> I thank authors for their detailed response and explanation. Much of my concerns are addressed and I am happy to improve my score. On reading other reviews, I should reiterate that the experimental evaluation is mediocre and can be improved.

---

### Author Response · Authors · 2021-11-23
**Revised version**

Dear reviewers,

We would like to thank you for your comments and suggestions. We believe that they have contributed to improving the quality of the paper a lot. We have uploaded a revised version on Openreview. The main changes are the following:

- The related work section has been made more concise.
- The part on equivariance has been significantly revised, in order to better highlight the novelty of our contribution.
- Additional explanations have been added for Top-n. We describe more precisely how the angle parameters are trained, how the number of reference points affects training and the differences with the Top-K pooling graph coarsening method of Gao & Ji (2019).
- A problem in the architecture for molecule generation has been fixed. Results are now significantly better for all set creation methods (up to 56% of valid and unique molecules instead of 38% in previous version), and they now outperform GraphVAE by a large margin. We also added a study of Top-n with a varying number of reference points on the synthetic set generation task. We however did not manage to run all the experiments suggested by the reviewers, such as the measure of a reconstruction error on a held-out set for the synthetic dataset.

---

### Decision · Program_Chairs · 2022-01-20

**Decision:**

Accept (Poster)

**Comment:**

Summary of the paper: This work considers the problem of generating sets and graphs conditioned on a latent representation (a.k.a. one-shot set generation) and makes two contributions.

First, it provides sufficient conditions for a learning algorithm to be able to handle permutation equivariance (the (F, l  ) equivariance). Second, it proposes Top-n, an approach for set-generation that builds on the set-generation method proposed by [1]. Top-n first generates an "angle vector" from a latent representation of the query vector, and then uses cosine similarity to select the closest n elements from a learnable reference set.

The authors compare their method with competing methods for set generation, including MLP-based generation, random-iid generation, and First-n creation. Their approach improves over these baselines on SetMNIST, synthetic molecule-like 3D structures, and the QM9 dataset.

Summary of discussions: The authors engaged extensively with the reviewers during the response period and were able to address significant reviewer concerns. While the reviewers are overall positive about the paper, it is expected that the authors will address some major concerns in the final camera-ready version. These include the lack of experiments on tasks used by [1], comparison with recursive methods, and discrepancies in TSPN results. Finally, the utility of the (F, l) equivariance is unclear, as most existing generative models already satisfy these conditions (as mentioned in authors' discussions with reviewer tgJk). Thus, the authors should adjust their claims accordingly and add necessary clarifications.